# TwinsFormer: Revisiting Inherent Dependencies via Two Interactive Components for Time Series Forecasting

## Abstract

Due to the remarkable ability to capture long-term dependencies, Transformer-based models have shown great potential in time series forecasting. However, real-world time series usually present intricate temporal patterns, making forecasting still challenging in many practical applications. To better grasp inherent dependencies, in this paper, we propose **TwinsFormer**, a Transformer-based framework utilizing two interactive components for time series forecasting. Unlike the mainstream paradigms of plain decomposition that train the model with two independent branches, we design an interactive strategy around the attention module and the feed-forward network to strengthen the dependencies via decomposed components. Specifically, we adopt dual streams to facilitate progressive and implicit information interactions for trend and seasonal components. For the seasonal stream, we feed the seasonal component to the attention module and feed-forward network with a subtraction mechanism. Meanwhile, we construct an auxiliary highway (without the attention module) for the trend stream by the supervision of seasonal signals. Finally, we incorporate the dual-stream outputs into a linear layer leading to the ultimate prediction. In this way, we can avoid the model overlooking inherent dependencies between different components for accurate forecasting. Our interactive strategy, albeit simple, can be adapted as a plug-and-play module to existing Transformer-based methods with negligible extra computational overhead. Extensive experiments on various real-world datasets show the superiority of TwinsFormer, which can outperform previous state-of-the-art methods in terms of both long-term and short-term forecasting performance.

## 1 Introduction

As a ubiquitous and paramount task in many real-world scenarios (e.g., weather (Wu et al., 2023b), energy (Yin et al., 2021), market (Granger & Newbold, 2014), and transportation (Yin et al., 2021)), time series forecasting has been explored with ongoing passion. Generally, time series forecasting aims to predict future temporal variations based on historical observations of time series, where the primary challenge is how to effectively capture temporal patterns from observed data. Benefiting from the advancements in deep learning, various representative models with well-designed architectures, such as MLP-based (Wang et al., 2024; Zeng et al., 2023; Li et al., 2023), CNN-based (Wang et al., 2023; Wu et al., 2023a; Liu et al., 2022a), and Transformer-based (Liu et al., 2024; Zhang & Yan, 2023; Zhou et al., 2022) methods, have been proposed to tackle time series forecasting tasks and demonstrate impressive performance. Since the complex and non-stationary nature of the real world or systems, the observed series usually involves multitudinous variations, such as increasing, decreasing, and fluctuating, making it still hard to grasp reliable and stable temporal dependencies.

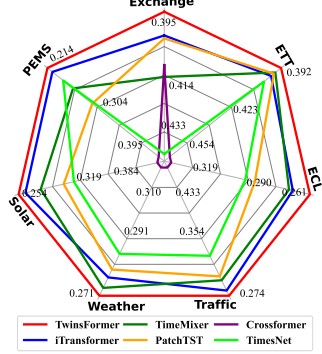

Figure 1: Performance of Twins-Former. Average results (MAE) are reported in Section 4.

To tackle intricate temporal patterns, series decomposition (Robert et al., 1990), utilizing the moving average kernel to smooth out short-term fluctuations or noise in the time series, has been involved

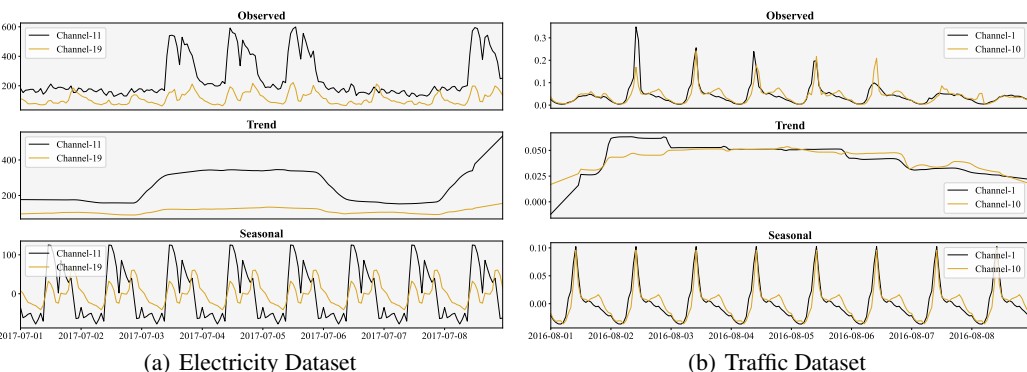

(a) Electricity Dataset         (b) Traffic Dataset

Figure 2: Trend-seasonal Decomposition. The top two subplots showcase the observed values of Electricity and Traffic from two different channels. The lower four subplots present the decomposed trend and seasonal components of the two datasets. Please zoom in for more details.

in deep models as a basic module. Empowered with various decomposition designs, existing methods (Wu et al., 2021; Zhou et al., 2022; Wang et al., 2023; Zeng et al., 2023) generally utilize two independent branches to highlight seasonal and trend properties separately, then combine the seasonal and trend representations for the final prediction. As seen in Figure 2, we decompose the raw signals (i.e., observed values) of two different channels on Electricity and Traffic datasets into trend and seasonal components for better understanding. Comparatively, the trend and seasonal components exhibit different but indispensable characteristics for the observed time series. More specifically, the former shows the overall variation while the latter presents the cyclical fluctuation. Since trend-seasonal decomposition is an untrainable linear transformation, the decomposed components obtained by the moving average kernel cannot directly reflect precise and intricate patterns for the observed time series. Taking channel 11 on the Electricity dataset as an example, significant variations and fluctuations of the observed series lag behind those of the trend and seasonal components. Such inconsistencies lead to the learned trend and seasonal representations by independent branches may not satisfy the temporal patterns of the observed series. Therefore, a more rational decomposition design should consider *the interactions between decomposed components to precisely unravel inherent dependencies for observed values.*

To fill this gap, we propose TwinsFormer, a Transformer-based framework that explicitly explores inherent dependencies via two interactive components for time series forecasting. **First**, we decompose the observed time series rather than the time series embeddings into trend and seasonal components better to capture the characteristics of the time series itself. **Second**, since the trend components reflect the long-term fluctuations of the time series, we only feed the seasonal components to the attention and feed-forward modules with a subtraction mechanism to alleviate redundant coding. **Most importantly**, we regard the outputs of the attention and feed-forward modules as supervision information to guide the model to capture the representation of the trend components. With our interactive design, TwinsFormer can successfully aggregate seasonal and trend information to learn inherent dependencies between different components for accurate forecasting. Experimentally, our proposed TwinsFormer achieves state-of-the-art performance on seven real-world forecasting scenarios shown in Figure 1 and effectively provides an interactive learning scheme for time series forecasting. The primary contributions are summarized as follows:

- We delve into the existing decomposition designs for time series forecasting and figure out that the interaction between different components is not explored: these designs simply learn separate representations for trend and seasonal components and overlook non-linear dependencies or significant noise levels among time series.

- We propose TwinsFormer, a Transformer-based framework (perhaps the first to our best knowledge) that explicitly explores inherent dependencies by learning implicit and progressive interactions between different components for time series forecasting.

- Extensive experimental results on 13 real-world benchmarks show the superiority of our TwinsFormer against previous state-of-the-art methods. Specifically, TwinsFormer ranks in the top 1 among 11 models on 18 out of the 22 average settings including various prediction lengths and metrics over long-term and short-term forecasting tasks.

## 2 RELATED WORK

### 2.1 DECOMPOSITIONS FOR TIME SERIES FORECASTING

Due to the capacity of the moving average kernel to smooth out short-term fluctuations or noise in the time series, Autoformer (Wu et al., 2021) initially proposed using the moving average kernel to decompose complex temporal variations into seasonal and trend components. Since then, trend-seasonal decomposition designs based on the moving average kernel have been frequently introduced in time series forecasting works. For instance, SCINet (Liu et al., 2022a) devises a downsample-convolve-interact architecture to extract dynamic temporal features at multiple resolutions with two sub-sequences. DLinear (Zeng et al., 2023) utilizes the series decomposition as the pre-processing before linear regression. MICN (Wang et al., 2023) adopts multi-scale branches to model the local and global context by decomposing input series into seasonal and trend terms while TimesNet (Wu et al., 2023a) designs a modular architecture to obtain decomposed components with Fourier Transform. More recently, TimeMixer (Wang et al., 2024) mixes multiscale decomposable components for time series forecasting. Due to the non-linear or non-stationary properties of time series, however, a rudimentary moving averaging kernel may inadequately capture precise trends, which impedes the model from learning inherent dependencies through two independent branches.

### 2.1.1 TRANSFORMERS FOR TIME SERIES FORECASTING

Benefiting from the ability to model long-term temporal patterns, Transformer-based methods (Li et al., 2019; Zhou et al., 2021; Liu et al., 2022b) have shown significant success in time series forecasting. Since the quadratic complexity and redundant coding for the self-attention mechanism, most existing Transformer-based approaches modify the attention module to reduce computational overhead. Representative works include Informer (Zhou et al., 2021) introducing ProbSparse self-attention and distilling techniques, Autoformer (Wu et al., 2021) incorporating series decomposition with an auto-correlation mechanism, FEDformer (Zhou et al., 2022) implementing the attention module with a Fourier-enhanced structure, etc. Without modifications to the Transformer, some other attempts pay attention to the inherent processing of time series, such as stationarity (Liu et al., 2022c; 2023), patching (Du et al., 2023), channel independence (Nie et al., 2023), and inverting operations (Liu et al., 2024), bringing consistently improved performance for time series forecasting. Besides, refurbishing Transformer in both aspects mentioned above, Crossformer Zhang & Yan (2023) introduces a two-stage-attention mechanism and dimension-segment-wise embedding strategy to capture cross-time and cross-variate dependencies.

Going beyond the designs in previous works, TwinsFormer devises an interactive dual-stream architecture without modifying the native components of Transformer. Moreover, we replace the observed series with trend and seasonal components, where the model can better learn inherent dependencies with their interactions. To the best of our knowledge, TwinsFormer is the first attempt to consider interactions between decomposed components on Transformer for time series forecasting.

## 3 TWINSFORMER

**Preliminary** Given the historical observation data $X = \{x_1, x_2, \cdots, x_M\} \in \mathbb{R}^{M \times N}$ with $M$ length look-back window and $N$ variates, the goal of multivariate time series forecasting is to predict the future time series $Y = \{x_{M+1}, \cdots, x_{M+\tau}\} \in \mathbb{R}^{\tau \times N}$ at next $\tau$ time steps ($\tau > 1$). Following the idea of decomposition (Robert et al., 1990; Wu et al., 2021), time series can be divided into trend and seasonal parts by the moving average kernel. For length-$M$ input series $X \in \mathbb{R}^{M \times N}$, the decomposition process can be formulated as

$$
\begin{aligned}
X_T &= AvgPool(Padding(X)), \\
X_S &= X - X_T,
\end{aligned}
\tag{1}
$$

where $X_T$ and $X_S$ denote the trend and seasonal components, respectively.

### 3.1 STRUCTURE OVERVIEW

Our proposed TwinsFormer illustrated on the left of Figure 3 adopts the encoder-only architecture, renovating Transformer to a dual-stream structure with two decomposed components. Before

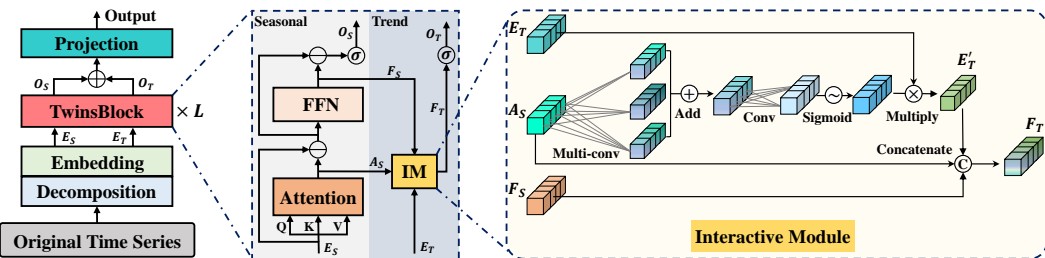

Figure 3: Overall framework of TwinsFormer.

embedding the time series, we decompose the observed series into trend (T) and seasonal (S) components in the channel dimension. Then, we feed seasonal embeddings $E_S$ to the attention module and feed-forward network (FFN) with a subtraction mechanism, while feeding trend embeddings $E_T$ to the interactive module with the supervision of seasonal information (i.e., $A_S$ and $F_s$). Finally, we aggregate seasonal and trend representations for the ultimate prediction.

**Embedding the decomposed series as tokens** For convenience, we denote $X_{m,:}$ as the simultaneously recorded values for all the variates at the $m$ time point, while $X_{:,n}$ as the whole time series of each variate indexed by $n$. Based on Equation 1, the trend and seasonal components of the time series can be formulated as $X_T = AvgPool(Padding(X_{:,n}))$ and $X_S = X - X_T$, respectively. Then, we utilize straightforward linear and dropout layers to create trend and seasonal embeddings with global covariates $X_{mark}$ as follows:

$$E_T = Dropout(Linear(Concat(X_T, X_{mark}))),$$
$$E_S = Dropout(Linear(Concat(X_S, X_{mark}))). \tag{2}$$

Note that, $Concat(\cdot)$ is used on the dataset containing $X_{mark}$ information and different components (i.e., $X_T$ and $X_S$) through separate liner layers in our experiments. In this way, we map decomposed series data $X_T, X_S \in \mathbb{R}^{N \times M}$ from the original space into a new space, where $E_T, E_S \in \mathbb{R}^{N \times D}$ and $D$ is the dimension of the linear layer.

**Learning interactions with our TwinsBlock** Unlike the existing Transformer variants that struggle to come up with efficient attention mechanisms to learn multivariate correlation, our Twins-Former incorporates interactive learning into Transformer block to explore the interactions between decomposed components for better inherent dependencies. Thus, a bundle of efficient attention mechanisms can be the plugins and our interactive strategy can promote the performance of existing Transformer variants on time series forecasting, which are evaluated in Table 4.

## 3.2 DUAL-STREAM DESIGN WITH INTERACTIVE MODULE

Keeping the original modules (i.e., the self-attention and feed-forward network (FFN) modules) of Transformer unchanged, our key design lies in the computationally efficient interactive module, which can guide the model to learn and aggregate the effective representations of trend and seasonal information. We provide the pseudo-code of our framework in Algorithm 1 of the Appendix.

**Seasonal Branch** Since the seasonal components obtained without the moving average kernel can better highlight the intrinsic characteristics of the time series data, we feed the seasonal embeddings to the attention and FFN modules to effectively capture the dependencies among the multivariates. Following the attention process of iTransformer (Liu et al., 2024), we regard $E_S \in \mathbb{R}^{N \times D}$ as $N$ $D$-dimension tokens and utilize queries, keys, and values $Q, K, V \in \mathbb{R}^{N \times d_k}$ to obtain the attention-weighted seasonal representations $A_S \in \mathbb{R}^{N \times D}$, where $d_k$ is the projected dimension

$$Q = E_s W_1 + b_1, K = E_s W_2 + b_2, V = E_s W_3 + b_3, \quad W_i \in \mathcal{R}^{d_k \times d_k}, b_i \in \mathcal{R}^{1 \times d_k},$$

$$A_S = Softmax(\frac{QK^T}{\sqrt{d_k}})V. \tag{3}$$

According to Equation 1, the seasonal components can be regarded as the residual part of the observed time series data. Intuitively, we adopt the idea of residual learning to implement a corrective strategy by subtracting the outputs of the Attention and FFN modules from the corresponding inputs.

The learning process can be formulated as follows:

$$H_1 = LayerNorm(E_S - A_S),$$
$$H_2 = H_1 - FFN(H_1).$$
(4)

**Trend Branch** Considering that untrainable moving average kernels lead to unreliable trend patterns, we fuse seasonal information to assist the learning of the trend branch through our interactive module (IM). On the one hand, attention-weighted $A_S$ well reflects the dependencies among multivariate, which can be converted into a coefficient matrix to update the trend embeddings. On the other hand, the signals discarded by the seasonal branch can be regarded as meaningful information to guide the representation of the trend embeddings. Our interactive module is illustrated on the right of Figure 3, we only use simple structures to train and update the trend branch network:

$$E'_T = Sigmoid(Conv_{1 \times 1}(\sum Multiconv(A_S))) * E_T,$$
$$F_T = Concate(A_S, F_S, E'_T),$$
(5)

where we concatenate $A_S$, $F_S$, and $E'_T$ in the embedding dimension and the kernel sizes of multi-scale convolutions are $1 \times 1$, $3 \times 3$, and $5 \times 5$, respectively.

**Gate Mechanism** Inspired by the inherent control of cells in RNNs (Zhao et al., 2017; Dey & Salem, 2017), we devise a gate mechanism $\sigma$ at the end of each block for both streams to autonomously regulate the pace of information transmission. The gate mechanism for both seasonal and trend streams can be formulated as

$$O_S = Sigmoid(Conv_1(H_2)) \cdot Conv_2(H_2),$$
$$O_T = Sigmoid(Conv_3(F_T)) \cdot Conv_4(F_T),$$
(6)

where $Conv_1, Conv_2, Conv_3$ and $Conv_4$ are four $1 \times 1$ convolution operations with different parameters. Taking the output of the former TwinsBlock as the input of the latter TwinsBlock, we stack $L$ TwinsBlocks to learn seasonal and trend representations, and then add them together through a linear projection for the ultimate predictive outcomes, i.e., $\{Y = Projection(O_S + O_T)\} \in \mathbb{R}^{\tau \times N}$.

**Rationality Analysis** Given historical time series data $X$, we can obtain its trend and seasonal components (i.e., $X_t$ and $X_s$) by the moving average kernel. For existing time series forecasting methods, we regard the models as $F(\cdot)$, while regarding the independent branches with decomposition designs as $f_t(\cdot)$ and $f_s(\cdot)$, then we can formulate the predictive outputs $Y$ as

$$Y = F(f_t(X_t) + f_s(X_s)), \quad where \ X = X_t + X_s.$$
(7)

Similarly, we definite the attention module, FFN, interactive module, and gate mechanism of Twins-Former as $g(\cdot)$, $h(\cdot)$, $\phi(\cdot)$, and $\sigma$ respectively. Then, the outcomes of TwinsFormer are:

$$Y = F(\overbrace{\sigma_t(\phi(\underbrace{[X_t, g(X_s), h(X_s - g(X_s))]}_{=X'_t}))}^{\text{interactive learning}} + \overbrace{\sigma_s(\underbrace{X_s - g(X_s) - h(X_s - g(X_s))}_{=X'_s})}^{\text{residual learning}}),$$
(8)

where $[\cdot]$ indicates concatenation operation. By omitting the constraints from various functions on variables and replacing '$[\cdot]$' with '+' operation, our interactive components can be simplified as

$$X'_s = X_s - X_1 - X_2, \quad X'_t = X_t + X_1 + X_2.$$
(9)

Then, the sum of our two interactive components can be expressed as

$$X = X'_s + X'_t = X_s - \cancel{X_1} - \cancel{X_2} + X_t + \cancel{X_1} + \cancel{X_2} = X_s + X_t.$$
(10)

Based on Equation (10), we can find that our interaction strategy perfectly fits the requirements of the decomposition design without bringing in redundant signals. Furthermore, we can elaborate on the practical implications of our TwinsFormer in mitigating the limitations of the trend-seasonal decomposition. Since the untrainable moving average kernel does not accurately capture the trend patterns, the decomposed trend and seasonal components are not completely disentangled. Twins-Former adopts a dual-stream interaction strategy to implicitly and progressively promote the decoupling of both components by using residual learning and interactive learning. Concretely, we filter out the coupled information (i.e., $X_1$ and $X_2$) from the seasonal components and compensate them to the trend components by some transformation mechanisms, so that we can learn more robust and reliable decomposed representations for accurate time series forecasting.

## 4 EXPERIMENTS

We conduct extensive experiments to evaluate the performance of TwinsFormer, covering long-term and short-term time series forecasting, including 13 real-world benchmarks and 10 well-acknowledged baselines. Moreover, we dive into the effectiveness and generality of the proposed framework to existing Transformer-based methods with indispensable ablation studies.

**Benchmarks** For long-term forecasting, we experiment on 9 well-established benchmarks: ETT (Zhou et al., 2021) (Electricity Transformer Temperature) datasets including ETTm1, ETTm2, ETTh1, and ETTh2, ECL (Wu et al., 2021) (Electricity Consuming Load), Exchange (Wu et al., 2021), Traffic (Wu et al., 2021), Weather (Wu et al., 2021) and Solar-energy (Lai et al., 2018) datasets. For short-term forecasting, we use 4 public traffic network PeMS (Liu et al., 2022a) datasets, namely PEMS03, PEMS04, PEMS07 and PEMS08. We follow standard protocols like (Wu et al., 2021; Liu et al., 2024) and split all datasets into training, validation and test sets in chronological order by the ratio of 6:2:2 for the ETT dataset and 7:1:2 for the other datasets. Detailed dataset descriptions are provided in Table 5 of the Appendix.

**Baselines** We compare TwinsFormer with 10 representative baselines, including 1) Transformer-based methods: iTransformer (Liu et al., 2024), PatchTST(Nie et al., 2023), Crossformer (Zhang & Yan, 2023), FEDformer (Zhou et al., 2022); 2)Linear-based methods: TimeMixer (Wang et al., 2024), DLinear (Zeng et al., 2023), TiDE (Das et al., 2023), and RLinear (Li et al., 2023); and 3) TCN-based methods: TimesNet Wu et al. (2023a), SCINet (Liu et al., 2022a).

**Implementation details** All the experiments are implemented in PyTorch (Paszke et al., 2019) and conducted on one NVIDIA 4090 24GB GPU. We use the L2 loss to train the model with the Adam (Kingma & Ba, 2015) optimizer, where the training process is early stopped within 10 epochs. Our TwinsBlock is applicable to Transformer-based architectures without introducing any additional hyperparameters. Following iTransformer (Liu et al., 2024), we use the Mean Square Error (MSE) and Mean Absolute Error (MAE) as the core metrics for the evaluation.

### 4.1 MAIN RESULTS

**Long-term forecasting** As shown in Table 1, TwinsFormer achieves leading performance on most benchmarks, covering various time series with different frequencies, variate numbers and real-world scenarios. For example, TwinsFormer outperforms iTransformer by a considerable margin, with a $6.2\%$ MSE reduction in ECL and a $5.1\%$ MSE reduction in Traffic. On Weather and Solar-energy datasets, although TimeMixer has a subtle advantage over TwinsFormer of $0.4\%$ and $4.8\%$ in MSE reduction, Twinsformer achieves lower MAE scores than TimeMixer by $1.1\%$ and $9.3\%$ reduction, respectively. It is worth noting that TwinsFormer exhibits better performance than TimeMixer among other datasets, which further highlights the superiority and robustness of TwinsFormer.

Table 1: Long-term forecasting results. The lookback length is set to $T = 96$ and all the results are averaged from all predictions $S \in \{96, 192, 336, 720\}$. Avg means further averaged by subsets. A lower MSE or MAE indicates a better prediction. See Table 6 in the Appendix for the full results.

| Models | TwinsFormer (Ours) | | iTransformer (2024) | | TimeMixer (2024) | | PatchTST (2023) | | RLinear (2023) | | Crossformer (2023) | | TiDE (2023) | | TimesNet (2023a) | | DLinear (2023) | | SCINet (2022a) | | FEDformer (2022) | |
|---|---|---|---|---|---|---|---|---|---|---|---|---|---|---|---|---|---|---|---|---|---|---|
| Metric | MSE | MAE | MSE | MAE | MSE | MAE | MSE | MAE | MSE | MAE | MSE | MAE | MSE | MAE | MSE | MAE | MSE | MAE | MSE | MAE | MSE | MAE |
| ETT (Avg) | **0.372** | **0.392** | 0.383 | 0.399 | 0.381 | 0.396 | 0.381 | 0.397 | 0.380 | **0.392** | 0.685 | 0.578 | 0.482 | 0.470 | 0.391 | 0.404 | 0.442 | 0.444 | 0.689 | 0.597 | 0.408 | 0.428 |
| ECL | **0.167** | **0.262** | 0.178 | 0.270 | 0.183 | 0.272 | 0.205 | 0.290 | 0.219 | 0.298 | 0.244 | 0.334 | 0.251 | 0.344 | 0.192 | 0.295 | 0.212 | 0.300 | 0.268 | 0.365 | 0.214 | 0.327 |
| Exchange | **0.346** | **0.395** | 0.360 | 0.403 | 0.380 | 0.417 | 0.367 | 0.404 | 0.378 | 0.417 | 0.940 | 0.707 | 0.370 | 0.413 | 0.416 | 0.443 | 0.354 | 0.414 | 0.750 | 0.626 | 0.519 | 0.429 |
| Traffic | **0.406** | **0.273** | 0.428 | 0.282 | 0.496 | 0.298 | 0.481 | 0.304 | 0.626 | 0.378 | 0.550 | 0.304 | 0.760 | 0.473 | 0.620 | 0.336 | 0.625 | 0.383 | 0.804 | 0.509 | 0.610 | 0.376 |
| Weather | 0.246 | **0.271** | 0.258 | 0.278 | **0.245** | 0.274 | 0.259 | 0.281 | 0.272 | 0.291 | 0.259 | 0.315 | 0.271 | 0.320 | 0.259 | 0.287 | 0.265 | 0.317 | 0.292 | 0.363 | 0.309 | 0.360 |
| Solar-energy | 0.227 | **0.254** | 0.233 | 0.262 | **0.216** | 0.280 | 0.270 | 0.307 | 0.369 | 0.356 | 0.641 | 0.639 | 0.347 | 0.417 | 0.301 | 0.319 | 0.330 | 0.401 | 0.282 | 0.375 | 0.291 | 0.381 |

**Short-term forecasting** TwinsFormer also performs well in short-term forecasting on PeMS datasets. Due to the complex spatiotemporal dependencies among citywide traffic networks in PeMS benchmarks, many advanced models degenerate a lot in this task. For instance, TimeMixer adopts the multiscale mixing architecture to model complex temporal variations, but its performance is not as good as iTransformer which simply tokenizes the embedding of time series in the variate dimension. By contrast, TwinsFormer learns the inherent dependencies from the interactions between

decomposed components, which can better capture accurate patterns for multivariate time series. Although SCINet obtains the best performance on the PEMS04 dataset using hierarchical sample convolution and interaction, it is inferior to TwinsFormer on other datasets. Remarkably, TwinsFormer achieves leading performance when averaging all the subsets, affirming the capacity of our interactive strategy in modeling complex temporal dynamics.

Table 2: Short-term forecasting results on PEMS datasets. The lookback length is set to $T = 96$ and all the results are averaged from all predictions $S \in \{12, 24, 48, 96\}$. A lower MSE or MAE indicates a better prediction. See Table 7 in the Appendix for the full results.

| Models | TwinsFormer (Ours) | | iTransformer (2024) | | TimeMixer (2024) | | PatchTST (2023) | | RLinear (2023) | | Crossformer (2023) | | TiDE (2023) | | TimesNet (2023a) | | DLinear (2023) | | SCINet (2022a) | | FEDformer (2022) | |
|--------|------|------|------|------|------|------|------|------|------|------|------|------|------|------|------|------|------|------|------|------|------|------|
| Metric | MSE | MAE | MSE | MAE | MSE | MAE | MSE | MAE | MSE | MAE | MSE | MAE | MSE | MAE | MSE | MAE | MSE | MAE | MSE | MAE | MSE | MAE |
| PEMS03 | **0.109** | **0.219** | 0.116 | 0.226 | 0.145 | 0.253 | 0.180 | 0.291 | 0.495 | 0.472 | 0.169 | 0.281 | 0.326 | 0.419 | 0.147 | 0.248 | 0.278 | 0.375 | 0.114 | 0.224 | 0.213 | 0.327 |
| PEMS04 | 0.111 | 0.219 | 0.121 | 0.232 | 0.162 | 0.268 | 0.195 | 0.307 | 0.526 | 0.491 | 0.209 | 0.314 | 0.353 | 0.437 | 0.129 | 0.241 | 0.295 | 0.388 | **0.092** | **0.202** | 0.231 | 0.337 |
| PEMS07 | **0.094** | **0.196** | 0.100 | 0.204 | 0.152 | 0.248 | 0.211 | 0.303 | 0.504 | 0.478 | 0.235 | 0.315 | 0.380 | 0.440 | 0.124 | 0.225 | 0.329 | 0.395 | 0.119 | 0.234 | 0.165 | 0.283 |
| PEMS08 | **0.133** | **0.222** | 0.151 | 0.234 | 0.209 | 0.296 | 0.280 | 0.321 | 0.529 | 0.487 | 0.268 | 0.307 | 0.441 | 0.464 | 0.193 | 0.271 | 0.379 | 0.416 | 0.158 | 0.244 | 0.286 | 0.358 |
| Avg | **0.112** | **0.214** | 0.122 | 0.224 | 0.167 | 0.266 | 0.217 | 0.305 | 0.514 | 0.482 | 0.220 | 0.304 | 0.375 | 0.440 | 0.148 | 0.246 | 0.320 | 0.394 | 0.121 | 0.222 | 0.224 | 0.327 |

## 4.2 ABLATION STUDIES

To verify the effectiveness of each main component of TwinsFormer, we provide indispensable ablation studies for every possible design on decomposition and interactions. To be concrete, we disable or replace certain designs as model variants and experiment on two long-term (i.e., ECL and Traffic) and two short-term forecasting (i.e., PEMS03 and PEMS07) datasets. As seen in Table 3, we conduct an insightful analysis of decomposition and interactions through the following observation.

Table 3: Ablation studies for TwinsFormer. We disable or replace each component of both decomposition and interactions over four datasets. ✓ and ✗ indicate with and without certain components, respectively. The average results of all predicted lengths are listed here. See Table 9 in the Appendix for complete ablation results.

| Design | Decomposition | Interactions | | | | | ECL | | Traffic | | PEMS03 | | PEMS07 | |
| | | $-$ | $E'_T$ | $A_S$ | $F_S$ | $\sigma$ | MSE | MAE | MSE | MAE | MSE | MAE | MSE | MAE |
|--------|------|------|------|------|------|------|------|------|------|------|------|------|------|------|
| **TwinsFormer** | ✓ | ✓ | ✓ | ✓ | ✓ | ✓ | **0.167** | **0.262** | **0.406** | **0.273** | **0.109** | **0.219** | **0.094** | **0.196** |
| ① | ✗ | ✓ | ✓ | ✓ | ✓ | ✓ | 0.176 | 0.272 | 0.417 | 0.282 | 0.116 | 0.226 | 0.102 | 0.204 |
| ② | swap | ✓ | ✓ | ✓ | ✓ | ✓ | 0.172 | 0.265 | 0.413 | 0.277 | 0.114 | 0.224 | 0.101 | 0.204 |
| ③ | ✓ | $+$ | ✓ | ✓ | ✓ | ✓ | 0.180 | 0.275 | 0.416 | 0.283 | 0.118 | 0.228 | 0.102 | 0.207 |
| ④ | ✓ | ✓ | ✗ | ✓ | ✓ | ✓ | 0.185 | 0.278 | 0.418 | 0.283 | 0.122 | 0.232 | 0.105 | 0.210 |
| ⑤ | ✓ | ✓ | ✓ | ✗ | ✓ | ✓ | 0.183 | 0.277 | 0.413 | 0.278 | 0.118 | 0.229 | 0.103 | 0.208 |
| ⑥ | ✓ | ✓ | ✓ | ✓ | ✗ | ✓ | 0.176 | 0.271 | 0.412 | 0.281 | 0.121 | 0.228 | 0.104 | 0.210 |
| ⑦ | ✓ | ✓ | ✓ | ✓ | ✓ | ✗ | 0.176 | 0.268 | 0.413 | 0.278 | 0.118 | 0.228 | 0.107 | 0.215 |

**Ablation on decomposition** Considering that the trend and seasonal components in the decomposition design are fed to different network branches, we disable the decomposition by using two original observed series as inputs (i.e., ①) and swap trend and seasonal components (i.e., ②) for ablation analysis. In ablation ① and ②, we can find significant decreases in forecasting performance for both long and short-term predictions, which demonstrates that our integration of the decomposition into Transformer architecture is reasonable and effective.

**Ablation on interactions** For the interactions, we verify the effectiveness by removing or replacing components gradually. In ablation ③, we replace the subtraction mechanism (i.e., $-$) with original addition skip connections (i.e., $+$), and the results on ③ show a decline in forecasting accuracy. This illustrates that decomposed components can better satisfy the requirements of Transformer architecture by using the subtraction mechanism. Meanwhile, the results in ③ further highlight the rationality of the decomposition design, which is consistent with the rationality analysis in Section 3.2. In ablations ④, ⑤, ⑥, and ⑦, we eliminate the impact of $E'_T$, $A_S$, $F_S$, and gate mechanism $\sigma$ for interactive learning, respectively. These four ablations all cause serious drops in forecasting performance, which indicates that all inputs for interactive learning can effectively boost the performance of TwinsFormer. The above observations highlight the substantial influence of our strategy using residual and interactive learning in Transformer architecture.

Table 4: Attention compatibility and performance promotion obtained by applying various efficient attention mechanisms to our interactive framework. The average results of all predicted lengths are listed here. See Table 10 in the Appendix for the full results.

| Models | | Transformer (2017) | | Informer (2021) | | Autoformer (2021) | | Flowformer (2022) | | Periodformer (2023) | |
|---|---|---|---|---|---|---|---|---|---|---|---|
| Metrics | | MSE | MAE | MSE | MAE | MSE | MAE | MSE | MAE | MSE | MAE |
| ECL | Original | 0.277 | 0.372 | 0.311 | 0.397 | 0.227 | 0.338 | 0.267 | 0.359 | 0.219 | 0.328 |
| | +Twins | **0.167** | **0.262** | **0.168** | **0.261** | **0.176** | **0.267** | **0.168** | **0.262** | **0.170** | **0.262** |
| | **Promotion** | **39.7%** | **29.6%** | **46.0%** | **43.7%** | **22.5%** | **21.0%** | **37.1%** | **27.0%** | **22.4%** | **20.1%** |
| Traffic | Original | 0.665 | 0.363 | 0.764 | 0.416 | 0.628 | 0.379 | 0.750 | 0.421 | 0.608 | 0.373 |
| | +Twins | **0.406** | **0.273** | **0.431** | **0.282** | **0.433** | **0.288** | **0.424** | **0.280** | **0.439** | **0.287** |
| | **Promotion** | **38.9%** | **24.8%** | **43.6%** | **32.2%** | **31.1%** | **24.0%** | **43.5%** | **33.5%** | **27.8%** | **23.1%** |
| PEMS03 | Original | 0.137 | 0.237 | 0.193 | 0.290 | 0.667 | 0.601 | 0.140 | 0.245 | 0.265 | 0.368 |
| | +Twins | **0.109** | **0.219** | **0.105** | **0.214** | **0.110** | **0.220** | **0.109** | **0.219** | **0.111** | **0.220** |
| | **Promotion** | **20.4%** | **7.6%** | **45.6%** | **26.2%** | **83.5%** | **63.4%** | **22.1%** | **10.6%** | **58.1%** | **40.2%** |
| PEMS07 | Original | 0.178 | 0.243 | 0.194 | 0.259 | 0.367 | 0.451 | 0.178 | 0.240 | 0.200 | 0.318 |
| | +Twins | **0.094** | **0.196** | **0.092** | **0.194** | **0.096** | **0.200** | **0.096** | **0.198** | **0.098** | **0.201** |
| | **Promotion** | **47.2%** | **19.3%** | **52.6%** | **25.1%** | **73.8%** | **55.7%** | **46.1%** | **17.5%** | **51.0%** | **36.8%** |

## 4.3 MODEL ANALYSIS

**Compatibility and promotion**  We evaluate TwinsFormer by applying our interactive strategy to original Transformer (Vaswani et al., 2017) and its variants, which generally address the quadratic complexity of the self-attention mechanism, including Informer (Zhou et al., 2021), Autoformer (Wu et al., 2021), Flowformer (Wu et al., 2022) and Periodformer (Liang et al., 2023). As seen in Table 4, our framework can be adapted to various attention mechanisms with promoted performance for Transformer-based forecasters. On the one hand, the performance under different attention mechanisms illustrates the favorable attention compatibility of TwinsFormer. On the other hand, the performance promotion for different Transformer-based architectures exhibits the superiority of TwinsFormer. Overall, it achieves averaged **28.4%** promotion on Transformer, **39.4%** on Informer, **46.9%** on Autoformer, **29.7%** on Flowformer and **34.9%** on Periodformer.

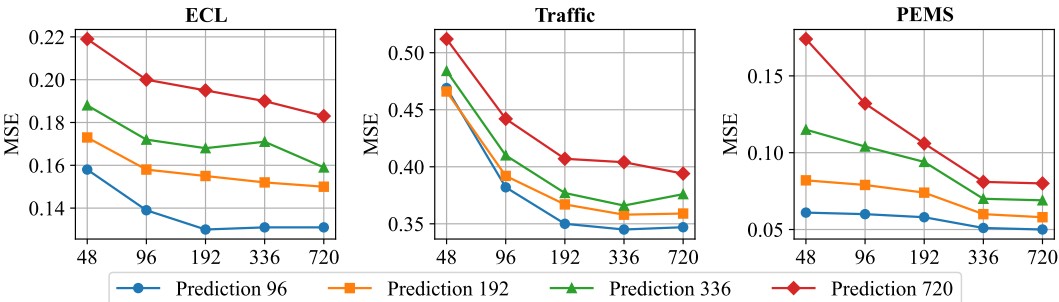

Figure 4: Forecasting performance with different lookback lengths on three datasets.

**Lookback length sensitivity**  As argued in (Zeng et al., 2023) and (Nie et al., 2023), most of the Transformer-based models will not improve the forecasting performance with an increasing lookback length due to the distracted attention on the longer input (Liu et al., 2024). However, our TwinsFormer reduces the MSE scores with enlarged historical information to be utilized, which is consistent with the theoretical analysis by statistical methods (Box & Jenkins, 1968). As seen in Figure 4, the forecasting results keep improving in most cases where the prediction length $S$ belongs to $\{96, 192, 336, 720\}$ as the receptive field increases. These improvements confirm that our TwinsFormer can effectively capture inherent dependencies from a longer lookback window.

**Visualization analysis**  To provide an intuitive understanding of the learned representations by our dual-stream framework, we visualize the multivariate correlations, the corresponding representations, and prediction results in Figure 5. It can be observed that the multivariate correlations learned by iTransformer are less distinct than those of TwinsFormer in the gold dashed box. Accordingly, the learned representations obtained by TwinsFormer have more abundant variate and temporal infor-

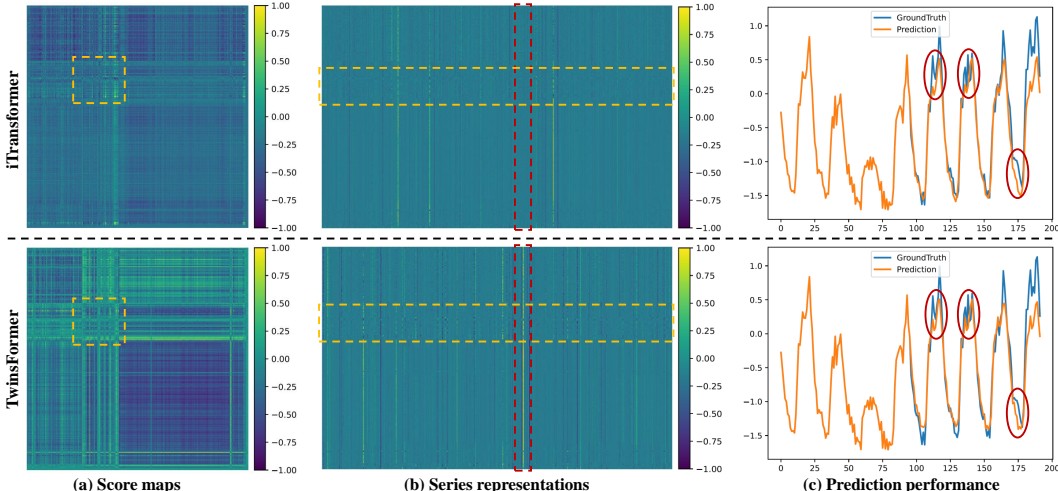

(a) Score maps      (b) Series representations      (c) Prediction performance

Figure 5: Analysis of multivariate correlations and series representations. Zoom in for more details.

mation than those of iTransformer, which we highlight with the gold and red dashed boxes in (b) of Figure 5. Consequently, TwinsFormer has better accuracy than iTransformer in forecasting performance, which is circled with a red line in the (c) part. Those observations indicate that TwinsFormer can better learn inherent dependencies among time series to achieve more accurate forecasting.

**Efficiency analysis** Our TwinsFormer is a Transformer-based architecture with dual-stream interactions, where the trend branch is composed of linear layers and sigmoid activation functions. Therefore, like other Transformer models, the main complexity of Twins-Former is $O(N^2)$, which comes from the seasonal branch with the attention module. Note that, the $N$ for TwinsFormer is related to the number of variates, while the $N$ for most Tansformer-based models is affected by the lookback length. Going further, the efficiency of TwinsFormer exceeds most Trans-

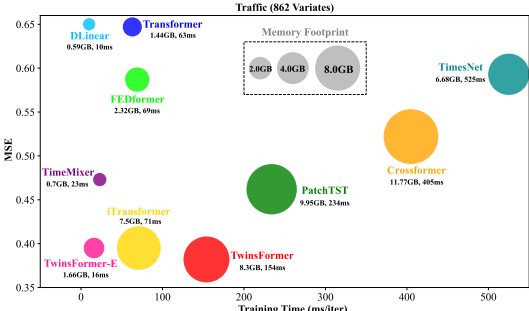

Figure 6: Model efficient comparison on Traffic.

former Variants in datasets with a relatively small number of variates (i.e., $N < 96$). Although the memory cost of TwinsFormer is not dominant when $N \gg 96$, we can choose only a part of the variates based on the correlations among variates to improve the training efficiency. As shown in Figure 6, TwinsFormer can obtain the best performance compared with 8 baselines on Traffic dataset, since the multivariate correlations can be well explored. Meanwhile, our efficient TwinsFormer-E (trained with 20% variates and prediction for all variates) can achieve comparable forecasting performance while substantially reducing the memory footprint.

## 5 CONCLUSION AND FUTURE WORK

Considering that the decomposition design can mine temporal patterns and the attention mechanism can capture multivariate correlations, we propose TwinsFormer, a Transformer-based framework revisiting inherent dependencies via two interactive branches for time series forecasting. Empowered with our interactive module, TwinsFormer handily incorporates the idea of decomposition into Transformer architectures and effectively learns time series representations. Experimentally, Twins-Former achieves state-of-the-art performances in both long-term and short-term forecasting tasks. Furthermore, the detailed visualization, ablations, and analysis illustrate the effectiveness and generality of our framework. Benefiting from the multivariate correlations learned with variate tokens, TwinsFormer demonstrates favorable run-time efficiency for high-dimension channel datasets. In the future, we will explore more efficient interaction design with decomposed components in the MLP architectures and analyze the performance of the interaction design in more time series tasks.

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

## A    IMPLEMENTATION DETAILS

**Benchmarks details**    We evaluate the performance of TwinsFormer compared with various baselines on 13 well-established benchmarks [1], which are detailed in Table 5.

Table 5: Detailed descriptions of benchmarks. Channel denotes the number of variates in each dataset. Prediction length points out four prediction settings. The dataset size is split in (Train, Validation, Test). Frequency denotes the sampling interval of time points.

| Tasks | Benchmarks | Channels | Prediction Length | Dataset Size | Frequency | Information |
|---|---|---|---|---|---|---|
| Long-term Forecasting | ETTm1 | 7 | {96, 192, 336, 720} | (34465, 11521, 11521) | 15min | Electricity |
| | ETTm2 | 7 | | (34465, 11521, 11521) | 15min | Electricity |
| | ETTh1 | 7 | | (8545, 2881, 2881) | Hourly | Electricity |
| | ETTh2 | 7 | | (8545, 2881, 2881) | Hourly | Electricity |
| | ECL | 321 | | (18317, 2633, 5261) | Hourly | Electricity |
| | Traffic | 862 | | (12185, 1757, 3509) | Hourly | Transportation |
| | Exchange | 8 | | (5120, 665, 1422) | Daily | Economy |
| | Weather | 21 | | (36792, 5271, 10540) | 10min | Weather |
| | Solar-energy | 137 | | (36601, 5161, 10417) | 10min | Electricity |
| Short-term Forecasting | PEMS03 | 358 | {12, 24, 48, 96} | (15617, 5135, 5135) | 5min | Transportation |
| | PEMS04 | 307 | | (10172, 3375, 3375) | 5min | Transportation |
| | PEMS07 | 883 | | (16911, 5622, 5622) | 5min | Transportation |
| | PEMS08 | 170 | | (10690, 3548, 3548) | 5min | Transportation |

**Metrics details**    Regarding evaluation metrics, we utilize the mean square error (MSE) and mean absolute error (MAE) for long-term and short-term forecasting:

$$\text{MSE} = \frac{1}{L}\sum_{i=1}^{L}(X_i - \hat{X}_i)^2, \qquad \text{MAE} = \sum_{i=1}^{L}|X_i - \hat{X}_i|,$$

where $X, \hat{X} \in \mathbb{R}^{L \times N}$ denote the ground truth and prediction results for $N$ variates in the future $L$ time steps. $|\cdot|$ means the absolute value operation.

**Algorithm details**    We provide the pseudo-code of TwinsFormer in Algorithm 1.

---

**Algorithm 1** Workflow of our TwinsFormer.

---

**Input:** Input lookback time series $X \in \mathbb{R}^{T \times N}$; Input length $T$, prediction length $L$, and variates number $N$; Token dimension $D$, TwinsBlock number $M$, and moving average kernel size $k$.

**Output:** The prediction results $\hat{X} \in \mathbb{R}^{L \times N}$.

1: ▷ Using the moving average kernel and padding operations to decompose time series.
2: $X_T = AvgPool(Padding(X)), X_S = X - X_T$      ▷ $X_T, X_S \in \mathbb{R}^{T \times N}$
3: ▷ Embedding series into variate tokens by Multi-layer Perceptron.
4: $E_T^0 = Embed_T(X_T.transpose), E_S^0 = Embed_S(X_S.transpose)$    ▷ $E_T^0, E_S^0 \in \mathbb{R}^{N \times D}$
5: ▷ Running through TwinsFormer blocks.
6: **for** $m$ in $\{1, \cdots, M\}$ **do**
7:   ▷ Self-attention mechanism and feed-forward network are applied for the seasonal branch.
8:   $E_S^{m-1} = LayerNorm(E_S^{m-1} - \text{Self-Attn}(E_S^{m-1}))$    ▷ $E_S^{m-1} \in \mathbb{R}^{N \times D}$
9:   $E_S^m = E_S^{m-1} - \text{Feed-Forward}(E_S^{m-1})$     ▷ $E_S^m \in \mathbb{R}^{N \times D}$
10:   ▷ Interactive module (IM) is utilized for the trend branch.
11:   $E_T^{m-1} = Sigmoid(Conv(\sum Multiconv(\text{Self-Attn}(E_S^{m-1})))) \cdot E_T^{m-1}$   ▷ $E_T^{m-1} \in \mathbb{R}^{N \times D}$
12:   $E_T^m = Concate([\text{Self-Attn}(E_S^{m-1}), \text{Feed-Forward}(E_S^{m-1}), E_T^{m-1}])$   ▷ $E_T^m \in \mathbb{R}^{N \times 3 \times D}$
13:   ▷ Adding gate mechanism to seasonal and trend branches.
14:   $E_S^m = LayerNorm(Sigmoid(Conv(E_S^m)) * E_S^m)$     ▷ $E_S^m \in \mathbb{R}^{N \times D}$
15:   $E_T^m = Sigmoid(Conv(E_T^m)) * E_T^m$      ▷ $E_T^m \in \mathbb{R}^{N \times D}$
16: **end for**
17: $\hat{X} = Projector(E_S^m + E_T^m)$       ▷ $\hat{X} \in \mathbb{R}^{N \times L}$
18: $\hat{X} = \hat{X}.transpose$        ▷ $\hat{X} \in \mathbb{R}^{L \times N}$
19: **return** $\hat{X}$

---

[1]All the datasets are publicly available at `https://github.com/thuml/iTransformer`

## B  BASELINE METHODS

We provide brief descriptions for the selected baselines as follows:

- **iTransformer** (Liu et al., 2024) is a Transformer-based model that captures multivariate correlations on the variate dimension for forecasting. The source code is available at `https://github.com/thuml/iTransformer`.

- **TimeMixer** (Wang et al., 2024) is an MLP-based predictor that adopts Past-Decomposable-Mixing and Future-Multipredictor-Mixing blocks to aggregate disentangled information. The source code is available at `https://github.com/kwuking/TimeMixer`.

- **PatchTST** (Nie et al., 2023) is a Transformer-based model that utilizes patching design and channel-independence to learn temporal patterns for forecasting. The source code is available at `https://github.com/yuqinie98/PatchTST`.

- **RLinear** (Li et al., 2023) is an MLP-based model that employs linear mapping with reversible normalization and independent channel operations for forecasting. The source code is available at `https://github.com/plumprc/RTSF`.

- **Crossformer** (Zhang & Yan, 2023) is a Transformer-based predictor capturing cross-time and cross-dimension dependencies with dimension-segment-wise embedding. The source code is available at `https://github.com/Thinklab-SJTU/Crossformer`.

- **TiDE** (Das et al., 2023) is an MLP-based predictor that handles covariates and non-linear dependencies with the dense encoder. The source code is available at `https://github.com/ZihangHLiu/TiDE`.

- **TimesNet** (Wu et al., 2023a) is a CNN-based model that ravels out the complex temporal variations into multiple intraperiod- and interperiod-variations for general time series analysis. The source code is available at `https://github.com/thuml/TimesNet`.

- **DLinear** (Zeng et al., 2023) is an MLP-based model that combines a decomposition scheme with two one-layer linear layers for forecasting. The source code is available at `https://github.com/cure-lab/LTSFLinear`.

- **SCINet** (Liu et al., 2022a) is a CNN-based model that conducts sample convolution and interaction for temporal modeling and forecasting. The source code is available at `https://github.com/cure-lab/SCINet`.

- **FEDformer** (Zhou et al., 2022) is a Transformer-based model that utilizes the seasonal-trend decomposition with frequency-enhanced blocks to learn temporal dependency for forecasting. The source code of FEDformer is available at `https://github.com/MAZiqing/FEDformer`.

- **Transformer** (Vaswani et al., 2017) utilizes self-attention mechanism to capture cross-time dependency for forecasting. The source code of Transformer is available at `https://github.com/zhouhaoyi/Informer2020`.

- **Informer** (Zhou et al., 2021) is a Transformer-based model using the ProbSparse self-attention to learn cross-time correlations for forecasting. The source code of Informer is available at `https://github.com/zhouhaoyi/Informer2020`.

- **Autoformer** (Wu et al., 2021) is a Transformer-based predictor introducing decomposition design and Auto-Correlation mechanism to capture temporal dependency. The source code of Autoformer is available at `https://github.com/thuml/Autoformer`.

- **Flowformer** (Wu et al., 2022) is a Transformer-based model that replaces the self-attention mechanism with the flow-attention. The source code is available at `https://github.com/thuml/Flowformer`.

- **Periodformer** (Liang et al., 2023) is a Transformer-based predictor capturing the periodicity of time series based on the relationship between different moments. The source code is available at `https://github.com/Anoise/Minusformer`.

To ensure the fairness of comparison, all the baselines are reproduced with the same repository, which is available at `https://github.com/thuml/Time-Series-Library`.

# C  FULL MAIN RESULTS

Due to the space limitation, we provide the full multivariate forecasting results here. Specifically, Table 6 contains the detailed results of all prediction lengths on 9 well-acknowledged benchmarks for long-term forecasting, while Table 7 includes the full short-term forecasting results on 4 challenging citywide traffic datasets.

Table 6: Full results of the long-term forecasting task. We compare extensive competitive models under different prediction lengths $S \in \{96, 192, 336, 720\}$. The input sequence length is set to 96 for all baselines. Avg means the average results from all four prediction lengths.

| Models | | TwinsFormer (Ours) | | iTransformer (2024) | | TimeMixer (2024) | | PatchTST (2023) | | RLinear (2023) | | Crossformer (2023) | | TiDE (2023) | | TimesNet (2023a) | | DLinear (2023) | | SCINet (2022a) | | FEDformer (2022) | |
|---|---|---|---|---|---|---|---|---|---|---|---|---|---|---|---|---|---|---|---|---|---|---|---|
| Metric | | MSE | MAE | MSE | MAE | MSE | MAE | MSE | MAE | MSE | MAE | MSE | MAE | MSE | MAE | MSE | MAE | MSE | MAE | MSE | MAE | MSE | MAE |
| ETTm1 | 96 | 0.325 | 0.364 | 0.334 | 0.368 | **0.320** | **0.355** | 0.329 | 0.367 | 0.355 | 0.376 | 0.404 | 0.426 | 0.364 | 0.387 | 0.338 | 0.375 | 0.345 | 0.372 | 0.418 | 0.438 | 0.379 | 0.419 |
| | 192 | 0.372 | 0.390 | 0.377 | 0.391 | **0.362** | **0.382** | 0.367 | 0.385 | 0.391 | 0.392 | 0.540 | 0.451 | 0.398 | 0.404 | 0.374 | 0.387 | 0.380 | 0.389 | 0.439 | 0.450 | 0.426 | 0.441 |
| | 336 | 0.406 | 0.412 | 0.426 | 0.420 | **0.396** | **0.406** | 0.399 | 0.410 | 0.424 | 0.415 | 0.532 | 0.515 | 0.428 | 0.425 | 0.410 | 0.411 | 0.413 | 0.413 | 0.490 | 0.485 | 0.445 | 0.459 |
| | 720 | 0.467 | 0.448 | 0.491 | 0.459 | 0.458 | 0.445 | **0.454** | **0.439** | 0.487 | 0.450 | 0.666 | 0.589 | 0.487 | 0.461 | 0.478 | 0.450 | 0.474 | 0.453 | 0.595 | 0.550 | 0.543 | 0.490 |
| | Avg | 0.393 | 0.404 | 0.407 | 0.410 | **0.384** | **0.397** | 0.387 | 0.400 | 0.414 | 0.407 | 0.513 | 0.496 | 0.419 | 0.419 | 0.400 | 0.406 | 0.403 | 0.407 | 0.485 | 0.481 | 0.448 | 0.452 |
| ETTm2 | 96 | **0.173** | **0.256** | 0.180 | 0.264 | 0.176 | 0.259 | 0.175 | 0.259 | 0.182 | 0.265 | 0.287 | 0.366 | 0.207 | 0.305 | 0.187 | 0.267 | 0.193 | 0.292 | 0.286 | 0.377 | 0.203 | 0.287 |
| | 192 | **0.239** | **0.300** | 0.250 | 0.309 | 0.242 | 0.303 | 0.241 | 0.302 | 0.246 | 0.304 | 0.414 | 0.492 | 0.290 | 0.364 | 0.249 | 0.309 | 0.284 | 0.362 | 0.399 | 0.445 | 0.269 | 0.328 |
| | 336 | **0.298** | **0.339** | 0.311 | 0.348 | 0.303 | 0.339 | 0.305 | 0.343 | 0.307 | 0.342 | 0.597 | 0.542 | 0.377 | 0.422 | 0.321 | 0.351 | 0.369 | 0.427 | 0.637 | 0.591 | 0.325 | 0.366 |
| | 720 | 0.397 | 0.397 | 0.412 | 0.407 | 0.396 | 0.399 | 0.402 | 0.400 | 0.407 | 0.398 | 1.730 | 1.042 | 0.558 | 0.524 | 0.408 | 0.403 | 0.554 | 0.522 | 0.960 | 0.735 | 0.421 | 0.415 |
| | Avg | **0.277** | **0.323** | 0.288 | 0.332 | 0.279 | 0.325 | 0.281 | 0.326 | 0.286 | 0.327 | 0.757 | 0.610 | 0.358 | 0.404 | 0.291 | 0.333 | 0.350 | 0.401 | 0.571 | 0.537 | 0.305 | 0.349 |
| ETTh1 | 96 | 0.385 | 0.401 | 0.386 | 0.405 | **0.384** | 0.400 | 0.414 | 0.419 | 0.386 | **0.395** | 0.423 | 0.448 | 0.479 | 0.464 | 0.384 | 0.402 | 0.386 | 0.400 | 0.654 | 0.599 | 0.376 | 0.419 |
| | 192 | 0.439 | 0.431 | 0.441 | 0.436 | 0.437 | 0.429 | 0.460 | 0.445 | **0.437** | **0.424** | 0.471 | 0.474 | 0.525 | 0.492 | 0.436 | 0.429 | 0.437 | 0.432 | 0.719 | 0.631 | 0.420 | 0.448 |
| | 336 | 0.480 | 0.452 | 0.487 | 0.458 | 0.472 | 0.446 | 0.501 | 0.466 | 0.479 | 0.446 | 0.570 | 0.546 | 0.565 | 0.515 | 0.491 | 0.469 | 0.481 | 0.459 | 0.778 | 0.659 | 0.459 | 0.465 |
| | 720 | **0.480** | **0.474** | 0.503 | 0.491 | 0.586 | 0.531 | 0.500 | 0.488 | 0.481 | 0.470 | 0.653 | 0.621 | 0.594 | 0.558 | 0.521 | 0.500 | 0.519 | 0.516 | 0.836 | 0.699 | 0.506 | 0.507 |
| | Avg | **0.446** | 0.440 | 0.454 | 0.447 | 0.470 | 0.451 | 0.469 | 0.454 | **0.446** | **0.434** | 0.529 | 0.522 | 0.541 | 0.507 | 0.458 | 0.450 | 0.456 | 0.452 | 0.747 | 0.647 | 0.440 | 0.460 |
| ETTh2 | 96 | 0.292 | 0.345 | 0.297 | 0.349 | 0.297 | 0.348 | 0.302 | 0.348 | **0.288** | **0.338** | 0.745 | 0.584 | 0.400 | 0.440 | 0.340 | 0.374 | 0.333 | 0.387 | 0.707 | 0.621 | 0.358 | 0.397 |
| | 192 | 0.375 | 0.395 | 0.380 | 0.400 | **0.369** | 0.392 | 0.388 | 0.400 | 0.374 | 0.390 | 0.877 | 0.656 | 0.528 | 0.509 | 0.402 | 0.414 | 0.477 | 0.476 | 0.860 | 0.689 | 0.429 | 0.439 |
| | 336 | 0.417 | 0.429 | 0.428 | 0.432 | 0.427 | 0.435 | 0.426 | 0.433 | **0.415** | **0.426** | 1.043 | 0.731 | 0.643 | 0.571 | 0.452 | 0.452 | 0.594 | 0.541 | 1.000 | 0.744 | 0.496 | 0.487 |
| | 720 | **0.406** | **0.430** | 0.427 | 0.445 | 0.462 | 0.463 | 0.431 | 0.446 | 0.420 | 0.440 | 1.104 | 0.763 | 0.874 | 0.679 | 0.462 | 0.468 | 0.831 | 0.657 | 1.249 | 0.838 | 0.463 | 0.474 |
| | Avg | **0.373** | **0.400** | 0.383 | 0.407 | 0.389 | 0.409 | 0.387 | 0.407 | 0.374 | 0.398 | 0.942 | 0.684 | 0.611 | 0.550 | 0.414 | 0.427 | 0.559 | 0.515 | 0.954 | 0.723 | 0.437 | 0.449 |
| ECL | 96 | **0.139** | **0.233** | 0.148 | 0.240 | 0.153 | 0.244 | 0.181 | 0.281 | 0.201 | 0.281 | 0.219 | 0.314 | 0.237 | 0.329 | 0.168 | 0.272 | 0.197 | 0.282 | 0.247 | 0.345 | 0.193 | 0.308 |
| | 192 | **0.158** | **0.252** | 0.162 | 0.253 | 0.168 | 0.259 | 0.188 | 0.274 | 0.201 | 0.283 | 0.231 | 0.322 | 0.236 | 0.330 | 0.184 | 0.289 | 0.196 | 0.285 | 0.257 | 0.355 | 0.201 | 0.315 |
| | 336 | **0.172** | **0.267** | 0.178 | 0.269 | 0.185 | 0.275 | 0.204 | 0.293 | 0.215 | 0.298 | 0.246 | 0.337 | 0.249 | 0.344 | 0.198 | 0.300 | 0.209 | 0.301 | 0.269 | 0.369 | 0.214 | 0.329 |
| | 720 | **0.200** | **0.293** | 0.225 | 0.317 | 0.227 | 0.312 | 0.246 | 0.324 | 0.257 | 0.331 | 0.280 | 0.363 | 0.284 | 0.373 | 0.220 | 0.320 | 0.245 | 0.333 | 0.299 | 0.390 | 0.246 | 0.355 |
| | Avg | **0.167** | **0.262** | 0.178 | 0.270 | 0.183 | 0.272 | 0.205 | 0.290 | 0.219 | 0.298 | 0.244 | 0.334 | 0.251 | 0.344 | 0.192 | 0.295 | 0.212 | 0.300 | 0.268 | 0.365 | 0.214 | 0.327 |
| Exchange | 96 | **0.081** | **0.200** | 0.086 | 0.206 | 0.099 | 0.218 | 0.088 | 0.205 | 0.093 | 0.217 | 0.256 | 0.367 | 0.094 | 0.218 | 0.107 | 0.234 | 0.088 | 0.218 | 0.267 | 0.396 | 0.148 | 0.278 |
| | 192 | **0.172** | **0.295** | 0.177 | 0.299 | 0.196 | 0.313 | 0.176 | 0.299 | 0.184 | 0.307 | 0.470 | 0.509 | 0.184 | 0.307 | 0.226 | 0.344 | 0.176 | 0.315 | 0.351 | 0.459 | 0.271 | 0.315 |
| | 336 | 0.320 | 0.409 | 0.331 | 0.417 | 0.359 | 0.432 | **0.301** | **0.397** | 0.351 | 0.432 | 1.268 | 0.883 | 0.349 | 0.431 | 0.367 | 0.448 | 0.313 | 0.427 | 1.324 | 0.853 | 0.460 | 0.427 |
| | 720 | **0.812** | **0.677** | 0.847 | 0.691 | 0.864 | 0.703 | 0.901 | 0.714 | 0.886 | 0.714 | 1.767 | 1.068 | 0.852 | 0.698 | 0.964 | 0.746 | 0.839 | 0.695 | 1.058 | 0.797 | 1.195 | 0.695 |
| | Avg | **0.346** | **0.395** | 0.360 | 0.403 | 0.380 | 0.417 | 0.367 | 0.404 | 0.378 | 0.417 | 0.940 | 0.707 | 0.370 | 0.413 | 0.416 | 0.443 | 0.354 | 0.414 | 0.750 | 0.626 | 0.519 | 0.429 |
| Traffic | 96 | **0.382** | **0.260** | 0.395 | 0.268 | 0.473 | 0.287 | 0.462 | 0.295 | 0.649 | 0.389 | 0.522 | 0.290 | 0.805 | 0.493 | 0.593 | 0.321 | 0.650 | 0.396 | 0.788 | 0.499 | 0.587 | 0.366 |
| | 192 | **0.392** | **0.267** | 0.417 | 0.276 | 0.486 | 0.294 | 0.466 | 0.296 | 0.601 | 0.366 | 0.530 | 0.293 | 0.756 | 0.474 | 0.617 | 0.336 | 0.598 | 0.370 | 0.789 | 0.505 | 0.604 | 0.373 |
| | 336 | **0.410** | **0.276** | 0.433 | 0.283 | 0.488 | 0.298 | 0.482 | 0.304 | 0.609 | 0.369 | 0.558 | 0.305 | 0.762 | 0.477 | 0.629 | 0.336 | 0.605 | 0.373 | 0.797 | 0.508 | 0.621 | 0.383 |
| | 720 | **0.442** | **0.292** | 0.467 | 0.302 | 0.536 | 0.314 | 0.514 | 0.322 | 0.647 | 0.387 | 0.589 | 0.328 | 0.719 | 0.449 | 0.640 | 0.350 | 0.645 | 0.394 | 0.841 | 0.523 | 0.626 | 0.382 |
| | Avg | **0.406** | **0.273** | 0.428 | 0.282 | 0.496 | 0.298 | 0.481 | 0.304 | 0.626 | 0.378 | 0.550 | 0.304 | 0.760 | 0.473 | 0.620 | 0.336 | 0.625 | 0.383 | 0.804 | 0.509 | 0.610 | 0.376 |
| Weather | 96 | **0.161** | **0.201** | 0.174 | 0.214 | 0.163 | 0.209 | 0.177 | 0.218 | 0.192 | 0.232 | 0.158 | 0.230 | 0.202 | 0.261 | 0.172 | 0.220 | 0.196 | 0.255 | 0.221 | 0.306 | 0.217 | 0.296 |
| | 192 | 0.211 | **0.248** | 0.221 | 0.254 | **0.209** | 0.252 | 0.225 | 0.259 | 0.240 | 0.271 | 0.206 | 0.277 | 0.242 | 0.298 | 0.219 | 0.261 | 0.237 | 0.296 | 0.261 | 0.340 | 0.276 | 0.336 |
| | 336 | 0.266 | 0.291 | 0.278 | 0.296 | **0.264** | **0.293** | 0.278 | 0.297 | 0.292 | 0.307 | 0.272 | 0.335 | 0.287 | 0.335 | 0.280 | 0.306 | 0.283 | 0.335 | 0.309 | 0.378 | 0.339 | 0.380 |
| | 720 | 0.347 | **0.343** | 0.358 | 0.347 | **0.345** | 0.345 | 0.354 | 0.348 | 0.364 | 0.353 | 0.398 | 0.418 | 0.351 | 0.386 | 0.365 | 0.359 | 0.345 | 0.381 | 0.377 | 0.427 | 0.403 | 0.428 |
| | Avg | 0.246 | **0.271** | 0.258 | 0.278 | **0.245** | 0.274 | 0.259 | 0.281 | 0.272 | 0.291 | 0.259 | 0.315 | 0.271 | 0.320 | 0.259 | 0.287 | 0.265 | 0.317 | 0.292 | 0.363 | 0.309 | 0.360 |
| Solar-Energy | 96 | 0.193 | **0.224** | 0.203 | 0.237 | **0.189** | 0.259 | 0.234 | 0.286 | 0.322 | 0.339 | 0.310 | 0.331 | 0.312 | 0.399 | 0.250 | 0.292 | 0.290 | 0.378 | 0.237 | 0.344 | 0.242 | 0.342 |
| | 192 | 0.223 | 0.250 | 0.233 | 0.261 | **0.222** | 0.283 | 0.267 | 0.310 | 0.359 | 0.356 | 0.734 | 0.725 | 0.339 | 0.416 | 0.296 | 0.318 | 0.320 | 0.398 | 0.280 | 0.380 | 0.285 | 0.380 |
| | 336 | 0.246 | 0.268 | 0.248 | 0.273 | **0.231** | 0.292 | 0.290 | 0.315 | 0.397 | 0.369 | 0.750 | 0.735 | 0.368 | 0.430 | 0.319 | 0.330 | 0.353 | 0.415 | 0.304 | 0.389 | 0.282 | 0.376 |
| | 720 | 0.245 | 0.272 | 0.249 | 0.275 | **0.223** | 0.285 | 0.289 | 0.317 | 0.397 | 0.356 | 0.769 | 0.765 | 0.370 | 0.425 | 0.338 | 0.337 | 0.356 | 0.413 | 0.308 | 0.388 | 0.357 | 0.427 |
| | Avg | 0.227 | **0.254** | 0.233 | 0.262 | **0.216** | 0.280 | 0.270 | 0.307 | 0.369 | 0.356 | 0.641 | 0.639 | 0.347 | 0.417 | 0.301 | 0.319 | 0.330 | 0.401 | 0.282 | 0.375 | 0.291 | 0.381 |
| 1st Count | | **22** | **30** | 0 | 0 | 18 | 6 | 2 | 2 | 4 | 9 | 0 | 0 | 0 | 0 | 0 | 0 | 0 | 0 | 0 | 0 | 0 | 0 |

TwinsFormer achieves the best forecasting performance among 11 models on various prediction horizons for both long-term and short-term forecasting tasks. To be concrete, TwinsFormer outperforms all the baselines on 52 out of the 90 settings including different prediction lengths and metrics over 9 long-term benchmarks. Meanwhile, TwinsFormer beats all the baselines on 29 out of the 40 settings of varying prediction lengths and metrics over 4 short-term datasets.

Table 7: Full results of the short-term forecasting task. We compare extensive competitive models under different prediction lengths $S \in \{12, 24, 48, 96\}$. The input sequence length is set to 96 for all baselines. Avg means the average results from all four prediction lengths.

| Models | | TwinsFormer (Ours) | | iTransformer (2024) | | TimeMixer (2024) | | PatchTST (2023) | | RLinear (2023) | | Crossformer (2023) | | TiDE (2023) | | TimesNet (2023a) | | DLinear (2023) | | SCINet (2022a) | | FEDformer (2022) | |
|---|---|---|---|---|---|---|---|---|---|---|---|---|---|---|---|---|---|---|---|---|---|---|---|
| Metric | | MSE | MAE | MSE | MAE | MSE | MAE | MSE | MAE | MSE | MAE | MSE | MAE | MSE | MAE | MSE | MAE | MSE | MAE | MSE | MAE | MSE | MAE |
| PEMS03 | 12 | **0.065** | **0.169** | 0.069 | 0.175 | 0.077 | 0.187 | 0.099 | 0.216 | 0.126 | 0.236 | 0.090 | 0.203 | 0.178 | 0.305 | 0.085 | 0.192 | 0.122 | 0.243 | 0.066 | 0.172 | 0.126 | 0.251 |
| | 24 | 0.086 | **0.196** | 0.097 | 0.208 | 0.112 | 0.224 | 0.121 | 0.240 | 0.246 | 0.334 | 0.121 | 0.240 | 0.257 | 0.371 | 0.118 | 0.223 | 0.201 | 0.317 | **0.085** | 0.198 | 0.149 | 0.275 |
| | 48 | **0.121** | **0.234** | 0.131 | 0.243 | 0.169 | 0.277 | 0.202 | 0.317 | 0.551 | 0.529 | 0.202 | 0.317 | 0.379 | 0.463 | 0.155 | 0.260 | 0.333 | 0.425 | 0.127 | 0.238 | 0.227 | 0.348 |
| | 96 | **0.165** | **0.276** | 0.168 | 0.279 | 0.22 | 0.322 | 0.262 | 0.367 | 1.057 | 0.787 | 0.262 | 0.367 | 0.490 | 0.539 | 0.228 | 0.317 | 0.457 | 0.515 | 0.178 | 0.287 | 0.348 | 0.434 |
| | Avg | **0.109** | **0.219** | 0.116 | 0.226 | 0.145 | 0.253 | 0.180 | 0.291 | 0.495 | 0.472 | 0.169 | 0.281 | 0.326 | 0.419 | 0.147 | 0.248 | 0.278 | 0.375 | 0.114 | 0.224 | 0.213 | 0.327 |
| PEMS04 | 12 | 0.077 | 0.181 | 0.081 | 0.188 | 0.092 | 0.203 | 0.105 | 0.224 | 0.138 | 0.252 | 0.098 | 0.218 | 0.219 | 0.340 | 0.087 | 0.195 | 0.148 | 0.272 | **0.073** | **0.177** | 0.138 | 0.262 |
| | 24 | 0.095 | 0.204 | 0.099 | 0.211 | 0.127 | 0.239 | 0.153 | 0.275 | 0.258 | 0.348 | 0.131 | 0.256 | 0.292 | 0.398 | 0.103 | 0.215 | 0.224 | 0.340 | **0.084** | **0.193** | 0.177 | 0.293 |
| | 48 | 0.120 | 0.231 | 0.133 | 0.247 | 0.188 | 0.294 | 0.229 | 0.339 | 0.572 | 0.544 | 0.205 | 0.326 | 0.409 | 0.478 | 0.136 | 0.250 | 0.355 | 0.437 | **0.099** | **0.211** | 0.270 | 0.368 |
| | 96 | 0.150 | 0.261 | 0.172 | 0.283 | 0.240 | 0.337 | 0.291 | 0.389 | 1.137 | 0.820 | 0.402 | 0.457 | 0.492 | 0.532 | 0.190 | 0.303 | 0.452 | 0.504 | **0.114** | **0.227** | 0.341 | 0.427 |
| | Avg | 0.111 | 0.219 | 0.121 | 0.232 | 0.162 | 0.268 | 0.195 | 0.307 | 0.526 | 0.491 | 0.209 | 0.314 | 0.353 | 0.437 | 0.129 | 0.241 | 0.295 | 0.388 | **0.092** | **0.202** | 0.231 | 0.337 |
| PEMS07 | 12 | **0.060** | **0.158** | 0.067 | 0.167 | 0.069 | 0.172 | 0.095 | 0.207 | 0.118 | 0.235 | 0.094 | 0.200 | 0.173 | 0.304 | 0.082 | 0.181 | 0.115 | 0.242 | 0.068 | 0.171 | 0.109 | 0.225 |
| | 24 | **0.079** | **0.181** | 0.086 | 0.189 | 0.106 | 0.212 | 0.150 | 0.262 | 0.242 | 0.341 | 0.139 | 0.247 | 0.271 | 0.383 | 0.101 | 0.204 | 0.210 | 0.329 | 0.119 | 0.225 | 0.125 | 0.244 |
| | 48 | **0.104** | **0.209** | 0.110 | 0.214 | 0.185 | 0.282 | 0.253 | 0.340 | 0.562 | 0.541 | 0.311 | 0.369 | 0.446 | 0.495 | 0.134 | 0.238 | 0.398 | 0.458 | 0.149 | 0.237 | 0.165 | 0.288 |
| | 96 | **0.132** | **0.236** | 0.138 | 0.244 | 0.246 | 0.327 | 0.346 | 0.404 | 1.096 | 0.795 | 0.396 | 0.442 | 0.628 | 0.577 | 0.181 | 0.279 | 0.594 | 0.553 | 0.141 | 0.304 | 0.262 | 0.376 |
| | Avg | **0.094** | **0.196** | 0.100 | 0.204 | 0.152 | 0.248 | 0.211 | 0.303 | 0.504 | 0.478 | 0.235 | 0.315 | 0.380 | 0.440 | 0.124 | 0.225 | 0.329 | 0.395 | 0.119 | 0.234 | 0.165 | 0.283 |
| PEMS08 | 12 | **0.075** | **0.174** | 0.080 | 0.183 | 0.097 | 0.205 | 0.168 | 0.232 | 0.133 | 0.247 | 0.165 | 0.214 | 0.227 | 0.343 | 0.112 | 0.212 | 0.154 | 0.276 | 0.087 | 0.184 | 0.173 | 0.273 |
| | 24 | **0.106** | **0.206** | 0.118 | 0.221 | 0.156 | 0.262 | 0.224 | 0.281 | 0.249 | 0.343 | 0.215 | 0.260 | 0.318 | 0.409 | 0.141 | 0.238 | 0.248 | 0.353 | 0.122 | 0.221 | 0.210 | 0.301 |
| | 48 | **0.167** | **0.258** | 0.186 | 0.265 | 0.269 | 0.345 | 0.321 | 0.354 | 0.569 | 0.544 | 0.315 | 0.355 | 0.497 | 0.510 | 0.198 | 0.283 | 0.440 | 0.470 | 0.189 | 0.270 | 0.320 | 0.394 |
| | 96 | **0.184** | **0.251** | 0.221 | 0.267 | 0.313 | 0.373 | 0.408 | 0.417 | 1.166 | 0.814 | 0.377 | 0.397 | 0.721 | 0.592 | 0.320 | 0.351 | 0.674 | 0.565 | 0.236 | 0.300 | 0.442 | 0.465 |
| | Avg | **0.133** | **0.222** | 0.151 | 0.234 | 0.209 | 0.296 | 0.280 | 0.321 | 0.529 | 0.487 | 0.268 | 0.307 | 0.441 | 0.464 | 0.193 | 0.271 | 0.379 | 0.416 | 0.158 | 0.244 | 0.286 | 0.358 |
| 1st Count | | **14** | **15** | 0 | 0 | 0 | 0 | 0 | 0 | 0 | 0 | 0 | 0 | 0 | 0 | 0 | 0 | 0 | 0 | 6 | 5 | 0 | 0 |

# D  ERROR BARS

We obtain the standard deviation of TwinsFormer performance by training the model with 5 different random seeds over 12 datasets. As seen in Table 8, the error bars of all the results are tiny, which exhibits that the performance of TwinsFormer is robust and reliable.

Table 8: Robustness of TwinsFormer performance obtained from 5 random seeds on 12 benchmarks.

| Dataset | | ETTm1 | | ETTm2 | | ETTh2 | | ECL | |
|---|---|---|---|---|---|---|---|---|---|
| Metrics | | MSE | MAE | MSE | MAE | MSE | MAE | MSE | MAE |
| | 96 | $0.325 \pm 0.001$ | $0.364 \pm 0.001$ | $0.173 \pm 0.001$ | $0.256 \pm 0.001$ | $0.292 \pm 0.001$ | $0.345 \pm 0.000$ | $0.139 \pm 0.000$ | $0.233 \pm 0.000$ |
| | 192 | $0.372 \pm 0.001$ | $0.390 \pm 0.002$ | $0.239 \pm 0.001$ | $0.300 \pm 0.000$ | $0.375 \pm 0.002$ | $0.395 \pm 0.001$ | $0.158 \pm 0.001$ | $0.252 \pm 0.001$ |
| | 336 | $0.406 \pm 0.002$ | $0.412 \pm 0.001$ | $0.298 \pm 0.000$ | $0.339 \pm 0.001$ | $0.417 \pm 0.004$ | $0.429 \pm 0.002$ | $0.172 \pm 0.002$ | $0.267 \pm 0.001$ |
| | 720 | $0.467 \pm 0.002$ | $0.448 \pm 0.003$ | $0.397 \pm 0.002$ | $0.397 \pm 0.001$ | $0.406 \pm 0.003$ | $0.430 \pm 0.001$ | $0.200 \pm 0.004$ | $0.293 \pm 0.002$ |

| Dataset | | Traffic | | Exchange | | Solar-Energy | | Weather | |
|---|---|---|---|---|---|---|---|---|---|
| Metrics | | MSE | MAE | MSE | MAE | MSE | MAE | MSE | MAE |
| | 96 | $0.382 \pm 0.001$ | $0.260 \pm 0.000$ | $0.081 \pm 0.001$ | $0.200 \pm 0.001$ | $0.193 \pm 0.002$ | $0.224 \pm 0.002$ | $0.161 \pm 0.000$ | $0.201 \pm 0.000$ |
| | 192 | $0.392 \pm 0.002$ | $0.267 \pm 0.001$ | $0.172 \pm 0.001$ | $0.295 \pm 0.001$ | $0.223 \pm 0.002$ | $0.250 \pm 0.002$ | $0.211 \pm 0.001$ | $0.248 \pm 0.001$ |
| | 336 | $0.410 \pm 0.003$ | $0.276 \pm 0.003$ | $0.320 \pm 0.002$ | $0.409 \pm 0.001$ | $0.246 \pm 0.000$ | $0.268 \pm 0.001$ | $0.266 \pm 0.002$ | $0.291 \pm 0.001$ |
| | 720 | $0.442 \pm 0.001$ | $0.292 \pm 0.001$ | $0.812 \pm 0.012$ | $0.677 \pm 0.005$ | $0.245 \pm 0.001$ | $0.272 \pm 0.001$ | $0.347 \pm 0.001$ | $0.343 \pm 0.001$ |

| Dataset | | PEMS03 | | PEMS04 | | PEMS07 | | PEMS08 | |
|---|---|---|---|---|---|---|---|---|---|
| Metrics | | MSE | MAE | MSE | MAE | MSE | MAE | MSE | MAE |
| | 12 | $0.065 \pm 0.000$ | $0.169 \pm 0.000$ | $0.077 \pm 0.003$ | $0.181 \pm 0.001$ | $0.060 \pm 0.000$ | $0.158 \pm 0.000$ | $0.075 \pm 0.000$ | $0.174 \pm 0.000$ |
| | 24 | $0.086 \pm 0.000$ | $0.196 \pm 0.000$ | $0.095 \pm 0.001$ | $0.204 \pm 0.000$ | $0.079 \pm 0.000$ | $0.181 \pm 0.000$ | $0.106 \pm 0.000$ | $0.206 \pm 0.000$ |
| | 48 | $0.121 \pm 0.001$ | $0.234 \pm 0.001$ | $0.120 \pm 0.002$ | $0.231 \pm 0.001$ | $0.104 \pm 0.001$ | $0.209 \pm 0.001$ | $0.167 \pm 0.001$ | $0.258 \pm 0.001$ |
| | 96 | $0.165 \pm 0.000$ | $0.276 \pm 0.001$ | $0.150 \pm 0.001$ | $0.261 \pm 0.000$ | $0.132 \pm 0.001$ | $0.236 \pm 0.001$ | $0.184 \pm 0.002$ | $0.251 \pm 0.001$ |

# E  FULL RESULTS FOR ABLATION STUDIES

To elaborate on the effectiveness of our TwinsFormer, we conduct detailed ablations covering disabling (✗), swapping (swap), and replacing components (+). Due to the page limitation, we provide detailed results and analysis here for ablation studies.

As shown in Table 8, we disable the decomposition and use the observed series as input for ①, where the results indicate that decomposed components are more effective for model performance than the

observed values as inputs. For ②, we swap the positions of the seasonal and trend components (i.e., feeding the trend components to the attention module while forcing the seasonal components as the input for the interactive module.), and the results show that the multivariate correlations captured by the attention module from the trend components are not effective as those obtained from the seasonal components. For ③, we replace the subtraction mechanism with the addition operation, in which the performance is inferior to TwinsFormer. This highlights that residual learning is more in line with the idea of decomposition design. As for ③, ④, ⑤, and ⑥, we disable $E_T'$, $A_S$, $F_S$, and $\sigma$ variants, respectively. The performance of the four cases is worse than that of the full model, indicating that each component is effective for the interactive module to capture inherent dependencies.

Table 9: Full ablation results on four benchmarks. We disable or replace each component of both decomposition and interactions for TwinsFormer. ✓ and ✗ indicate with and without certain components, respectively. H1, H2, H3, and H4 denote different prediction lengths, where prediction length belongs to $S \in \{96, 192, 336, 720\}$ for ECL and Traffic, while $S \in \{12, 24, 48, 96\}$ for PEMS03 and PEMS07. Avg means the average results of all predicted lengths.

| Design | Decomposition | Interactions − | $E_T'$ | $A_S$ | $F_S$ | $\sigma$ | Prediction lengths | ECL MSE | ECL MAE | Traffic MSE | Traffic MAE | PEMS03 MSE | PEMS03 MAE | PEMS07 MSE | PEMS07 MAE |
|---|---|---|---|---|---|---|---|---|---|---|---|---|---|---|---|
| **TwinsFormer** | ✓ | ✓ | ✓ | ✓ | ✓ | ✓ | H1 | **0.139** | **0.233** | **0.382** | **0.260** | **0.065** | **0.169** | **0.060** | **0.158** |
|  |  |  |  |  |  |  | H2 | **0.158** | **0.252** | **0.392** | **0.267** | **0.086** | **0.196** | **0.079** | **0.181** |
|  |  |  |  |  |  |  | H3 | **0.172** | **0.267** | **0.410** | **0.276** | **0.121** | **0.234** | **0.104** | **0.209** |
|  |  |  |  |  |  |  | H4 | **0.200** | **0.293** | **0.442** | **0.292** | **0.165** | **0.276** | **0.132** | **0.236** |
|  |  |  |  |  |  |  | Avg | **0.167** | **0.262** | **0.406** | **0.273** | **0.109** | **0.219** | **0.094** | **0.196** |
| ① | ✗ | ✓ | ✓ | ✓ | ✓ | ✓ | H1 | 0.142 | 0.245 | 0.388 | 0.265 | 0.070 | 0.176 | 0.065 | 0.162 |
|  |  |  |  |  |  |  | H2 | 0.160 | 0.258 | 0.405 | 0.278 | 0.093 | 0.204 | 0.083 | 0.185 |
|  |  |  |  |  |  |  | H3 | 0.178 | 0.272 | 0.420 | 0.284 | 0.133 | 0.242 | 0.112 | 0.214 |
|  |  |  |  |  |  |  | H4 | 0.224 | 0.312 | 0.456 | 0.301 | 0.169 | 0.283 | 0.146 | 0.254 |
|  |  |  |  |  |  |  | Avg | 0.176 | 0.272 | 0.417 | 0.282 | 0.116 | 0.226 | 0.102 | 0.204 |
| ② | swap | ✓ | ✓ | ✓ | ✓ | ✓ | H1 | 0.140 | 0.238 | 0.389 | 0.263 | 0.069 | 0.175 | 0.061 | 0.160 |
|  |  |  |  |  |  |  | H2 | 0.161 | 0.256 | 0.399 | 0.271 | 0.092 | 0.202 | 0.082 | 0.187 |
|  |  |  |  |  |  |  | H3 | 0.175 | 0.268 | 0.414 | 0.279 | 0.128 | 0.239 | 0.115 | 0.217 |
|  |  |  |  |  |  |  | H4 | 0.210 | 0.298 | 0.448 | 0.296 | 0.167 | 0.281 | 0.145 | 0.251 |
|  |  |  |  |  |  |  | Avg | 0.172 | 0.265 | 0.413 | 0.277 | 0.114 | 0.224 | 0.101 | 0.204 |
| ③ | ✓ | + | ✓ | ✓ | ✓ | ✓ | H1 | 0.145 | 0.248 | 0.388 | 0.268 | 0.072 | 0.177 | 0.063 | 0.161 |
|  |  |  |  |  |  |  | H2 | 0.168 | 0.263 | 0.402 | 0.271 | 0.095 | 0.204 | 0.081 | 0.186 |
|  |  |  |  |  |  |  | H3 | 0.182 | 0.276 | 0.414 | 0.289 | 0.135 | 0.243 | 0.115 | 0.224 |
|  |  |  |  |  |  |  | H4 | 0.223 | 0.311 | 0.458 | 0.304 | 0.171 | 0.286 | 0.147 | 0.255 |
|  |  |  |  |  |  |  | Avg | 0.180 | 0.275 | 0.416 | 0.283 | 0.118 | 0.228 | 0.102 | 0.207 |
| ④ | ✓ | ✓ | ✗ | ✓ | ✓ | ✓ | H1 | 0.149 | 0.248 | 0.392 | 0.269 | 0.074 | 0.178 | 0.064 | 0.163 |
|  |  |  |  |  |  |  | H2 | 0.172 | 0.265 | 0.407 | 0.275 | 0.099 | 0.207 | 0.083 | 0.189 |
|  |  |  |  |  |  |  | H3 | 0.188 | 0.282 | 0.419 | 0.284 | 0.138 | 0.251 | 0.124 | 0.231 |
|  |  |  |  |  |  |  | H4 | 0.229 | 0.318 | 0.452 | 0.303 | 0.176 | 0.290 | 0.149 | 0.257 |
|  |  |  |  |  |  |  | Avg | 0.185 | 0.278 | 0.418 | 0.283 | 0.122 | 0.232 | 0.122 | 0.210 |
| ⑤ | ✓ | ✓ | ✓ | ✗ | ✓ | ✓ | H1 | 0.148 | 0.251 | 0.389 | 0.264 | 0.073 | 0.179 | 0.065 | 0.163 |
|  |  |  |  |  |  |  | H2 | 0.172 | 0.263 | 0.399 | 0.270 | 0.095 | 0.205 | 0.084 | 0.192 |
|  |  |  |  |  |  |  | H3 | 0.188 | 0.281 | 0.413 | 0.278 | 0.131 | 0.247 | 0.119 | 0.225 |
|  |  |  |  |  |  |  | H4 | 0.222 | 0.311 | 0.449 | 0.300 | 0.173 | 0.284 | 0.143 | 0.251 |
|  |  |  |  |  |  |  | Avg | 0.183 | 0.277 | 0.413 | 0.278 | 0.118 | 0.229 | 0.103 | 0.208 |
| ⑥ | ✓ | ✓ | ✓ | ✓ | ✗ | ✓ | H1 | 0.142 | 0.243 | 0.385 | 0.265 | 0.072 | 0.175 | 0.066 | 0.165 |
|  |  |  |  |  |  |  | H2 | 0.163 | 0.257 | 0.396 | 0.273 | 0.096 | 0.203 | 0.084 | 0.194 |
|  |  |  |  |  |  |  | H3 | 0.184 | 0.279 | 0.417 | 0.285 | 0.139 | 0.244 | 0.120 | 0.223 |
|  |  |  |  |  |  |  | H4 | 0.215 | 0.303 | 0.451 | 0.302 | 0.177 | 0.288 | 0.147 | 0.259 |
|  |  |  |  |  |  |  | Avg | 0.176 | 0.271 | 0.412 | 0.281 | 0.121 | 0.228 | 0.104 | 0.210 |
| ⑦ | ✓ | ✓ | ✓ | ✓ | ✓ | ✗ | H1 | 0.141 | 0.236 | 0.388 | 0.268 | 0.069 | 0.174 | 0.079 | 0.193 |
|  |  |  |  |  |  |  | H2 | 0.160 | 0.255 | 0.404 | 0.271 | 0.094 | 0.204 | 0.084 | 0.190 |
|  |  |  |  |  |  |  | H3 | 0.178 | 0.270 | 0.414 | 0.279 | 0.132 | 0.245 | 0.116 | 0.223 |
|  |  |  |  |  |  |  | H4 | 0.223 | 0.310 | 0.445 | 0.294 | 0.178 | 0.289 | 0.147 | 0.254 |
|  |  |  |  |  |  |  | Avg | 0.176 | 0.268 | 0.413 | 0.278 | 0.118 | 0.228 | 0.107 | 0.215 |

In a nutshell, our decomposition design and interactive module are of importance for multivariate time series forecasting. We take the intrinsic modules (i.e., attention module and feed-forward network) as the cornerstone and utilize residual and interactive learning to better learn temporal and multivariate correlations, thereby further enhancing the model capacity for forecasting performance.

Table 10: Full results of promotion and attention compatibility on five Transformer-based models for both long-term and short-term forecasting tasks.

| Models | | | Transformer (2017) | | Informer (2021) | | Autoformer (2021) | | Flowformer (2022) | | Periodformer (2023) | |
|---|---|---|---|---|---|---|---|---|---|---|---|---|
| Metrics | | | MSE | MAE | MSE | MAE | MSE | MAE | MSE | MAE | MSE | MAE |
| ECL | Original | 96 | 0.260 | 0.358 | 0.274 | 0.368 | 0.201 | 0.317 | 0.215 | 0.320 | 0.180 | 0.294 |
| | | 192 | 0.266 | 0.367 | 0.296 | 0.386 | 0.222 | 0.334 | 0.259 | 0.355 | 0.193 | 0.307 |
| | | 336 | 0.280 | 0.375 | 0.300 | 0.394 | 0.231 | 0.338 | 0.296 | 0.383 | 0.208 | 0.321 |
| | | 720 | 0.302 | 0.386 | 0.373 | 0.439 | 0.254 | 0.361 | 0.296 | 0.380 | 0.293 | 0.390 |
| | | Avg | 0.277 | 0.372 | 0.311 | 0.397 | 0.227 | 0.338 | 0.267 | 0.359 | 0.219 | 0.328 |
| | +Twins | 96 | 0.139 | 0.233 | 0.138 | 0.232 | 0.151 | 0.242 | 0.140 | 0.234 | 0.142 | 0.235 |
| | | 192 | 0.158 | 0.252 | 0.156 | 0.248 | 0.161 | 0.253 | 0.159 | 0.252 | 0.157 | 0.249 |
| | | 336 | 0.172 | 0.267 | 0.170 | 0.264 | 0.176 | 0.269 | 0.171 | 0.267 | 0.173 | 0.267 |
| | | 720 | 0.200 | 0.293 | 0.208 | 0.301 | 0.215 | 0.303 | 0.201 | 0.293 | 0.207 | 0.298 |
| | | Avg | 0.167 | 0.262 | 0.168 | 0.261 | 0.176 | 0.267 | 0.168 | 0.262 | 0.170 | 0.262 |
| Traffic | Original | 96 | 0.647 | 0.357 | 0.719 | 0.391 | 0.613 | 0.388 | 0.691 | 0.393 | 0.562 | 0.343 |
| | | 192 | 0.649 | 0.356 | 0.696 | 0.397 | 0.616 | 0.382 | 0.729 | 0.419 | 0.587 | 0.356 |
| | | 336 | 0.667 | 0.364 | 0.777 | 0.420 | 0.622 | 0.337 | 0.756 | 0.423 | 0.612 | 0.370 |
| | | 720 | 0.697 | 0.376 | 0.864 | 0.472 | 0.660 | 0.408 | 0.825 | 0.449 | 0.672 | 0.423 |
| | | Avg | 0.665 | 0.363 | 0.764 | 0.416 | 0.628 | 0.379 | 0.750 | 0.421 | 0.608 | 0.373 |
| | +Twins | 96 | 0.382 | 0.260 | 0.397 | 0.266 | 0.400 | 0.268 | 0.392 | 0.264 | 0.408 | 0.273 |
| | | 192 | 0.392 | 0.267 | 0.420 | 0.276 | 0.420 | 0.276 | 0.412 | 0.273 | 0.426 | 0.280 |
| | | 336 | 0.410 | 0.276 | 0.436 | 0.284 | 0.439 | 0.284 | 0.428 | 0.281 | 0.443 | 0.288 |
| | | 720 | 0.442 | 0.292 | 0.471 | 0.303 | 0.473 | 0.305 | 0.465 | 0.301 | 0.477 | 0.307 |
| | | Avg | 0.406 | 0.273 | 0.431 | 0.282 | 0.433 | 0.288 | 0.424 | 0.280 | 0.439 | 0.287 |
| PEMS03 | Original | 96 | 0.105 | 0.205 | 0.202 | 0.293 | 0.272 | 0.385 | 0.105 | 0.207 | 0.128 | 0.257 |
| | | 192 | 0.121 | 0.222 | 0.173 | 0.277 | 0.334 | 0.440 | 0.127 | 0.229 | 0.173 | 0.306 |
| | | 336 | 0.145 | 0.248 | 0.186 | 0.286 | 1.032 | 0.782 | 0.155 | 0.257 | 0.268 | 0.384 |
| | | 720 | 0.175 | 0.274 | 0.211 | 0.304 | 1.031 | 0.796 | 0.173 | 0.287 | 0.490 | 0.524 |
| | | Avg | 0.137 | 0.237 | 0.193 | 0.290 | 0.667 | 0.601 | 0.140 | 0.245 | 0.265 | 0.368 |
| | +Twins | 96 | 0.065 | 0.169 | 0.064 | 0.167 | 0.066 | 0.171 | 0.065 | 0.169 | 0.065 | 0.169 |
| | | 192 | 0.086 | 0.196 | 0.084 | 0.193 | 0.087 | 0.196 | 0.085 | 0.195 | 0.086 | 0.195 |
| | | 336 | 0.121 | 0.234 | 0.115 | 0.228 | 0.122 | 0.235 | 0.120 | 0.233 | 0.121 | 0.234 |
| | | 720 | 0.165 | 0.276 | 0.157 | 0.269 | 0.165 | 0.277 | 0.164 | 0.277 | 0.170 | 0.280 |
| | | Avg | 0.109 | 0.219 | 0.105 | 0.214 | 0.110 | 0.220 | 0.109 | 0.219 | 0.111 | 0.220 |
| PEMS07 | Original | 96 | 0.173 | 0.235 | 0.189 | 0.255 | 0.199 | 0.336 | 0.174 | 0.236 | 0.117 | 0.238 |
| | | 192 | 0.174 | 0.238 | 0.193 | 0.258 | 0.323 | 0.420 | 0.173 | 0.236 | 0.147 | 0.275 |
| | | 336 | 0.181 | 0.245 | 0.196 | 0.261 | 0.390 | 0.470 | 0.183 | 0.244 | 0.226 | 0.353 |
| | | 720 | 0.185 | 0.252 | 0.196 | 0.260 | 0.554 | 0.578 | 0.180 | 0.245 | 0.309 | 0.407 |
| | | Avg | 0.178 | 0.243 | 0.194 | 0.259 | 0.367 | 0.451 | 0.178 | 0.240 | 0.200 | 0.318 |
| | +Twins | 96 | 0.060 | 0.158 | 0.060 | 0.157 | 0.061 | 0.160 | 0.060 | 0.158 | 0.061 | 0.159 |
| | | 192 | 0.079 | 0.181 | 0.078 | 0.181 | 0.082 | 0.186 | 0.079 | 0.181 | 0.080 | 0.184 |
| | | 336 | 0.104 | 0.209 | 0.103 | 0.207 | 0.108 | 0.214 | 0.105 | 0.209 | 0.109 | 0.215 |
| | | 720 | 0.132 | 0.236 | 0.128 | 0.232 | 0.134 | 0.239 | 0.140 | 0.245 | 0.140 | 0.245 |
| | | Avg | 0.094 | 0.196 | 0.092 | 0.194 | 0.096 | 0.200 | 0.096 | 0.198 | 0.098 | 0.201 |

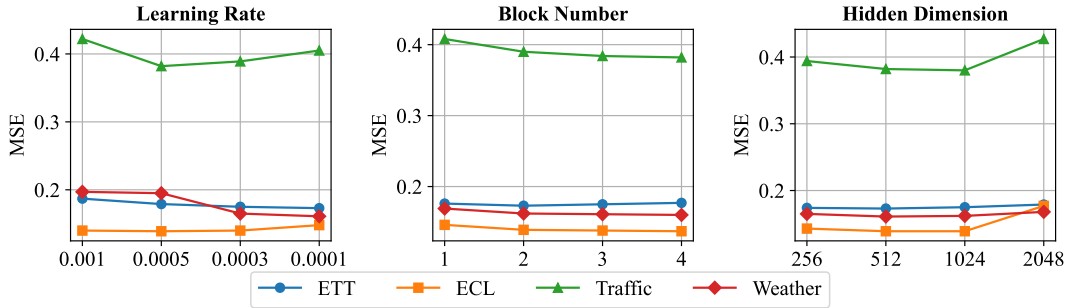

Figure 7: Hyperparameter sensitivity concerning the learning rate, the number of Twinsblock, and the hidden dimension of variate tokens. The results are recorded with the input length $T = 96$ and the prediction length $S = 96$ on four benchmarks.

## F  EXTRA RESULTS FOR MODEL ANALYSIS

We apply the proposed Twinsblocks to Transformer and its variants and compare the performance with the original results. Specifically, we regard Transformer (Vaswani et al., 2017), Informer (Zhou

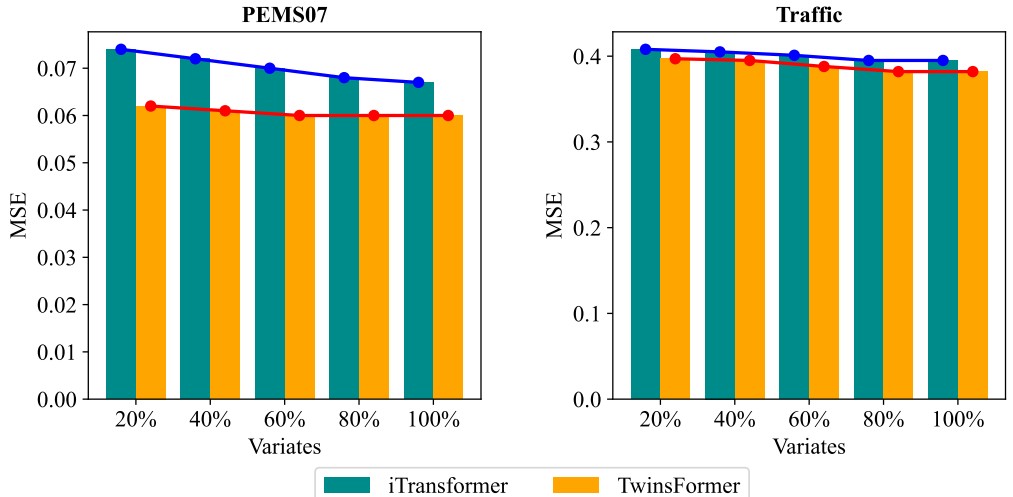

Figure 8: Performance of generation on unseen variates, comparing with iTransformer on PEMS07 and Traffic datasets. We train the model with $\{20\%, 40\%, 60\%, 80\%, 100\%\}$ variates, and evaluate the performance by forecasting all variates.

et al., 2021), Autoformer (Wu et al., 2021), Flowformer (Wu et al., 2022), and Periodformer (Liang et al., 2023) as original models, and treat various combinations of attention mechanisms and our interactive strategy as Twins variants. As shown in Table 8, our framework has robust performance under various attention mechanisms for the same benchmark and obtains better forecasting accuracy than original Transformer-based variants when using the same attention mechanism.

We evaluate the hyperparameter sensitivity of TwinsFormer in terms of the learning rate, the number of Twinsblock, and the hidden dimension of variate tokens. As shown in Figure 7, The performance fluctuates under different hyperparameter settings. We can observe that the learning rate, as the most common hyperparameter, should be carefully selected for different datasets. In most cases, increasing the number of the Twinsblock tends to strengthen the model performance, especially in datasets with numerous varieties. For scenarios involving many attributes, the forecasting performance will decrease when the hidden dimension of variate tokens is larger than 1024.

As seen in Figure 8, we set 5 variate proportions to explore the model generalization performance for short-term and long-term forecasting. From the figure, we can observe that the prediction performance is better as the number of variate increases for both forecasting tasks. Comparatively, the performance of TwinsFormer is better than that of iTransformer under any variate proportion, which demonstrates that our method has a more powerful generalization ability than iTransformer. Notably, TwinsFormer can be naturally trained with 20% variates and learn transferable representations to achieve favorable forecasting on all varieties.

## G SHOWCASES

### G.1 VISUALIZATION OF REPRESENTATIONS

To better understand the representations learned by our model, we visualize the learned multivariate correlations by the pre-Softmax scores. Following (Liu et al., 2024), we normalize each variate token on its feature dimension and reveal the variate-wise correlation by the whole score map $\mathbf{A} \in \mathbb{R}^{N \times N}$ among $N$ paired variate tokens. Meanwhile, we visualize the trend, seasonal, and weighted representations by normalized operation. As shown in Figure 9, we visualize the learned representations of iTranformer (Liu et al., 2024) for comparison to illustrate the superiority of our method. Based on Figure 9, we can observe that the multivariate correlations learned by TwinsFormer are clearer than those of iTransformer. Consequently, the learned representation via seasonal and trend branches can capture more effective information than iTranformer without decomposition design. Taking the representation in Figure 9 for example, (b), (c) and (d) contain more obvious information in the gold dashed box, and (a) has no variations within the red dashed boxes in the corresponding positions of (b), (c) and (d). Such differences show that TwinsFormer can better capture inherent dependencies of time series than iTransformer for forecasting tasks.

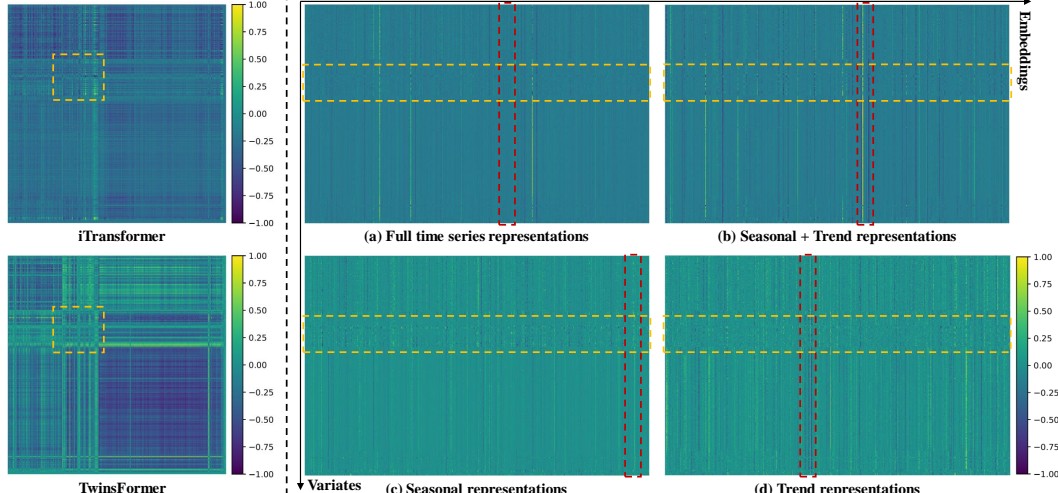

Figure 9: Analysis of multivariate correlations and series representations, comparing with iTransformer on ECL dataset. The left part of the dashed line denotes the multivariate correlations learned by iTransformer and TwinsFormer. Full series representations are obtained using iTransformer, and the rest are extracted from TwinsFormer. Zoom in for more details.

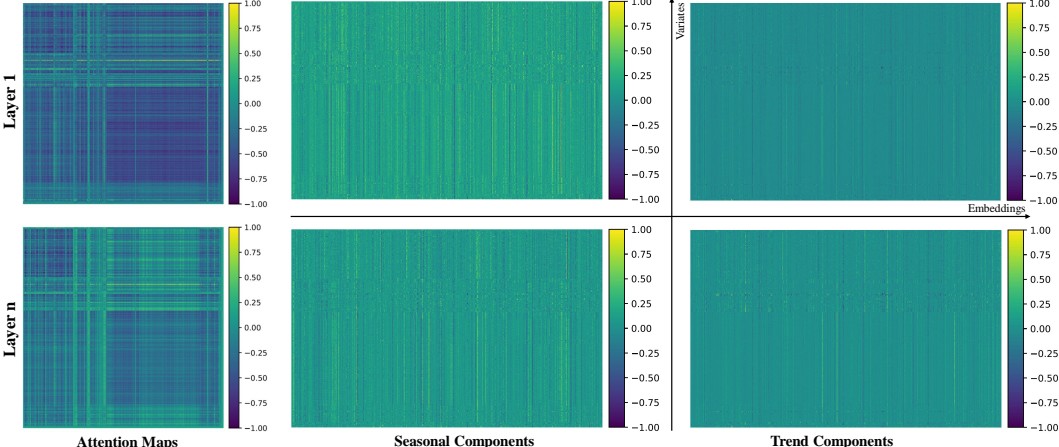

Figure 10: Analysis of multivariate correlations and series representations in different TwinsBlock on ECL dataset. The attention map and decomposed components of Layer 1 and n are obviously different, which indicates that our dual-stream framework can learn the interactions between seasonal and trend components, and thus better capture the multivariate dependencies.

In Figure 10, we provide the visualization of the attention maps and the decomposed components in different layers. Naturally, the inputs fed to the first TwinsBlock are the initial decomposed components obtained by the moving average kernel, while the inputs to the last TwinsBlock are decomposed components learned by our dual-stream interaction structure. Accordingly, the attention map of Layer 1 captures the multivariate correlations among the initial seasonal components, while the attention map of Layer n obtains the multivariate correlations among the learned seasonal components. The attention map and decomposed components of Layer 1 and n are obviously different, which indicates that our dual-stream framework can learn the interactions between seasonal and trend components, and thus better capture the multivariate dependencies.

In Figure 11-12, we present a detailed decomposed comparison of the moving average kernel and our interactions. Note that, the decomposed components by the moving average kernel can be regarded as initial decomposed components, while the decomposed components by our interactions are learned from our TwinsFormer. From these two figures, we can observe that the learned components are different from the initial components, where especially the learned seasonal component

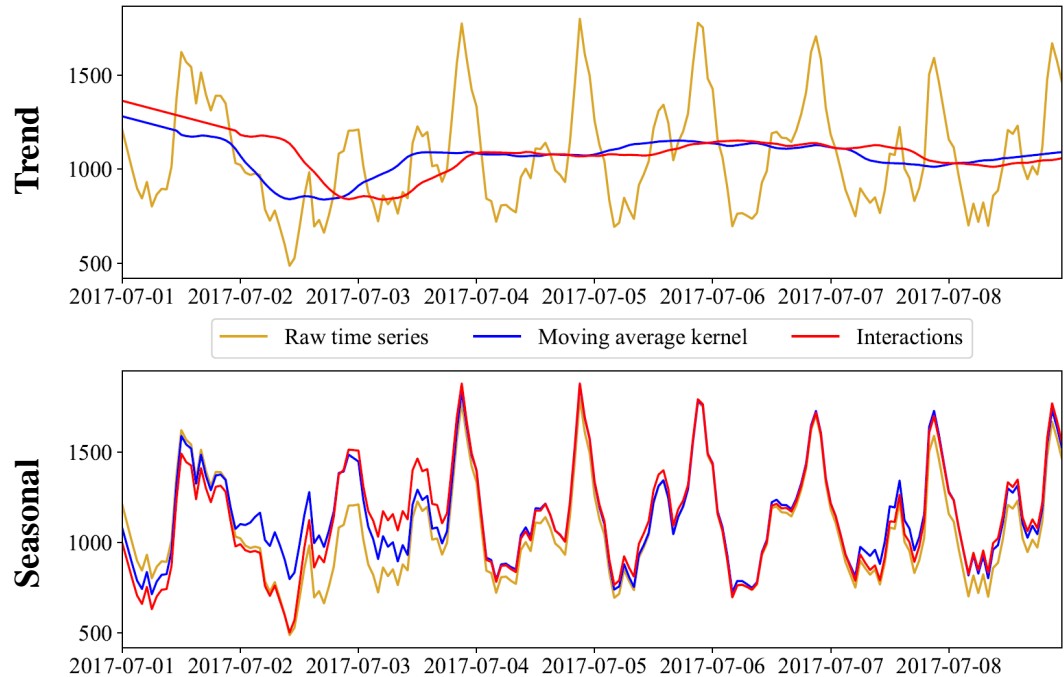

Figure 11: Trend-seasonal decomposition results obtained by the moving average kernel and our interactions on ECL. To better compare the variations of raw time series with the seasonal components, we translate the seasonal components on the vertical axis.

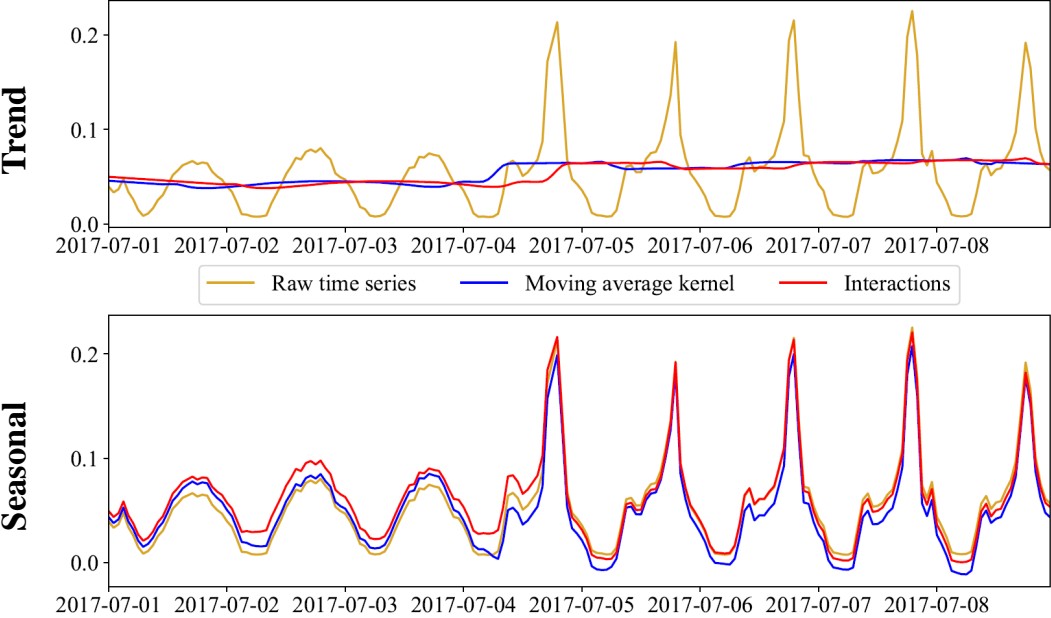

Figure 12: Trend-Seasonal Decomposition Results obtained by the moving average kernel and our interactions on Traffic. To better compare the variations of raw time series with the seasonal components, we translate the seasonal components on the vertical axis.

can better reflect the variations of the original time series than the initial seasonal component. Such variations fully demonstrate that our dual-stream framework can better capture the inherent dependencies of time series.

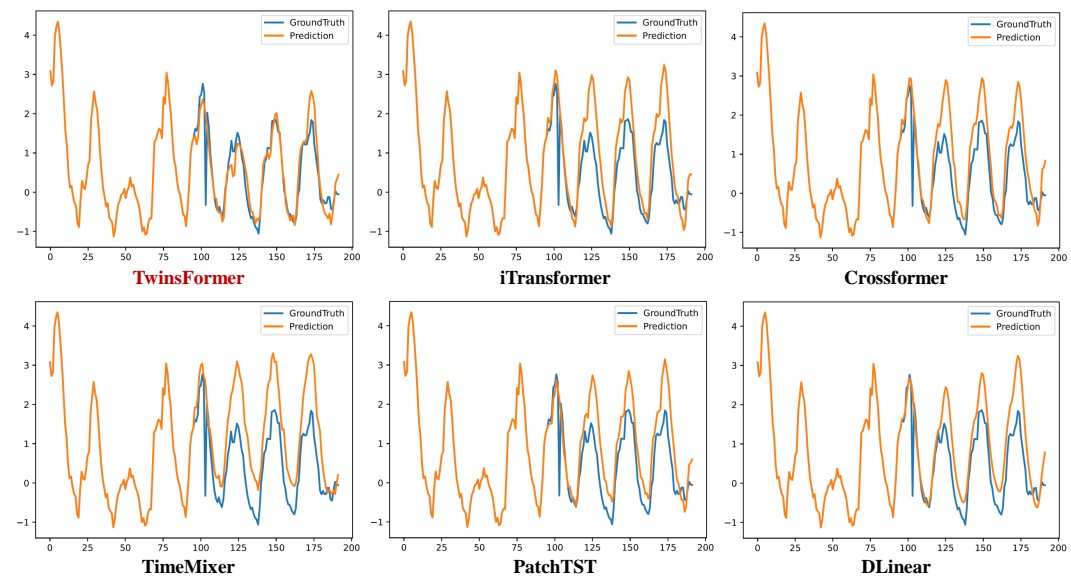

Figure 13: ECL prediction cases among different models under the input-96-predict-96 setting.

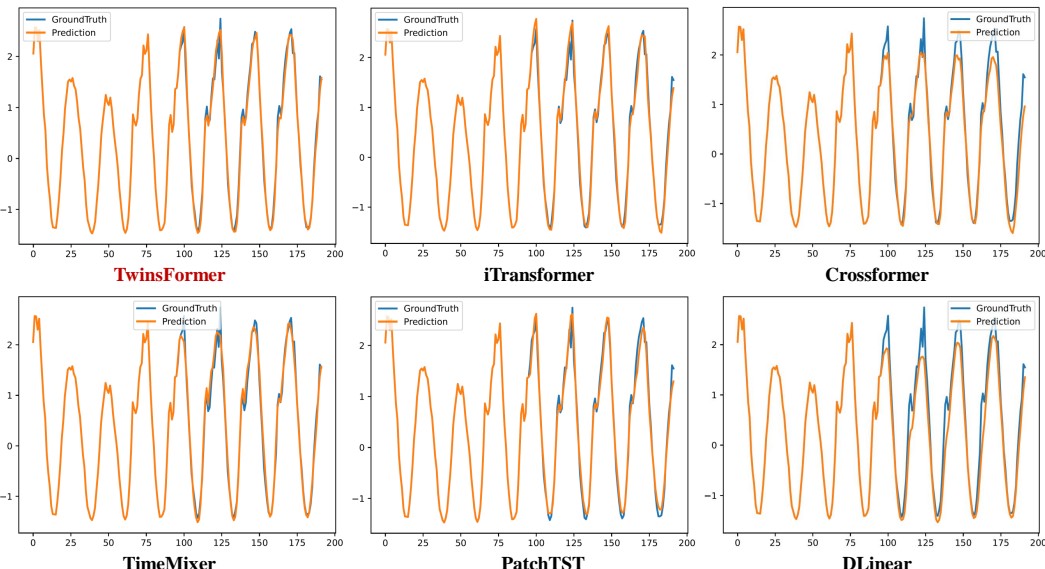

Figure 14: Traffic prediction cases among different models under the input-96-predict-96 setting.

## G.2 VISUALIZATION OF PREDICTION RESULTS

For clarity and comparison among different models, we present supplementary forecasting show-cases on four representative benchmarks in Figure 13, 14, 15, and 16. To be concrete, we provide prediction cases for TwinsFormer, iTransformer (Liu et al., 2024), TimeMixer (Wang et al., 2024), PatchTST (Nie et al., 2023), Crossformer (Zhang & Yan, 2023), and DLinear (Zeng et al., 2023) over ECL, Traffic, Weather, and PEMS07 datasets. Among these models, TwinsFormer exhibits superior forecasting performance with the most precise future series variations. Note that, all the qualitative results of different models are obtained with a fixed input length $T = 96$ and forecasting horizon $S = 96$ over four datasets.

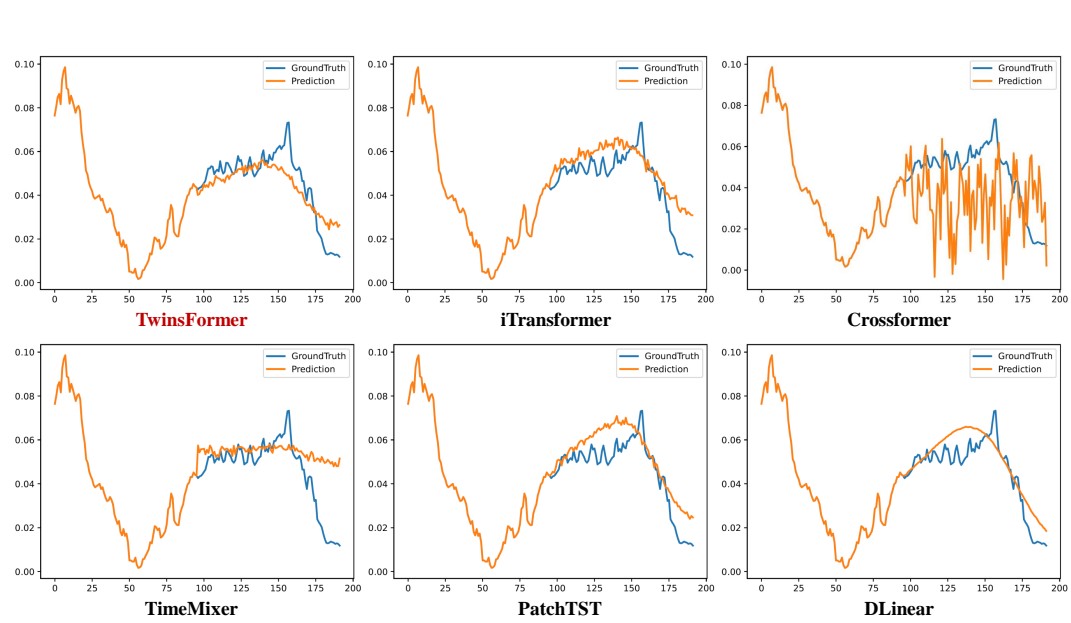

Figure 15: Weather prediction cases among different models under the input-96-predict-96 setting.

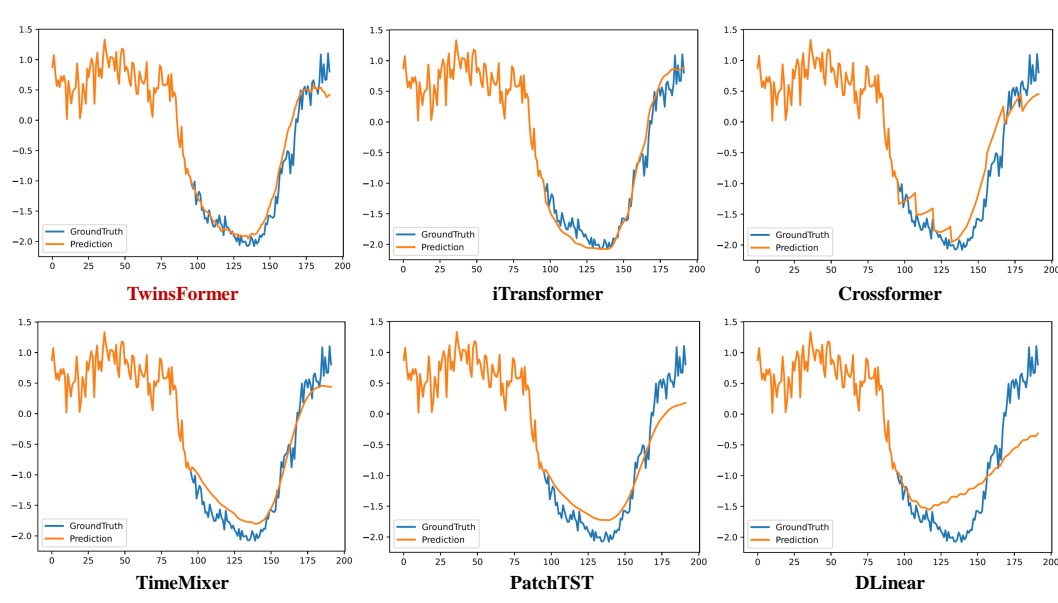

Figure 16: PEMS07 prediction cases among different models under the input-96-predict-96 setting.

