# OpenReview forum: "TwinsFormer: Revisiting Inherent Dependencies via Two Interactive Components for Time Series Forecasting"
_ICLR.cc/2025/Conference — Submitted to ICLR 2025_

### Official Review · Reviewer_pkJh · 2024-11-02

**Soundness:** 2
**Presentation:** 2
**Contribution:** 2
**Rating:** 5
**Confidence:** 3

**Summary:**

The paper proposes a method to improve transformer based time series forecasting. A dual stream architecture is proposed, where the multi-variate time series is decomposed into trend and seasonality components. Then the learned embedding for each of these components are then combined through an interactive module, to give the final combined representation. To my understanding, the key differentiator from existing works that do decomposition, is that, Twinsformer learns a better interaction between the decomposed components leading to a better downstream forecasting performance. Empirical performance is quite detailed on long term and short term forecasting tasks, and compared to the baselines, the performance is quite strong. Ablation studies appear comprehensive.

**Strengths:**

Originality & Significance
- strong empirical performance
- interesting idea to address interaction among different components, but not very novel

**Weaknesses:**

- the motivation is not very convincing
- "inherent dependencies" are not well defined
- lines 79-82, motivation on channel 11 is not clear, how does this make a case for "inherent dependencies", or what that is?
- lines 191-194 - it seems the goal has changed to address the problem of limitations in "multi-variate correlation" which is addressed by Twinsformer - makes it further unclear what the goal is
- many of the design choices also do not seem to have a clear motivation, and seem heuristic, and are post-hoc justified in ablation rather than a clear reason for selection

**Questions:**

- Please help clarify the motivations and the key problem being addressed through the proposed module, as it seems rather ad-hoc and heuristic

---

> ### Author Response · Authors · 2024-11-21
> **Response to Reviewer pkJh**
>
> Thanks for your time and effort in reviewing our paper. We are pleased to see that you identified the experimental results. However, we guess that there are some misunderstandings about the novelty of our work, which we try to clarify in our following responses to specific questions and concerns:
>
> Generally, __time series forecasting__ aims to predict future temporal variations based on historical values of time series, where the primary challenge is how to effectively capture __inherent dependencies__ from historical data.
> Inherent dependencies refer to the __correlations and patterns__ within the data that are crucial for accurate prediction, such as temporal dependencies, seasonal and trend patterns, autocorrelation, multivariate correlation, and so on.
> More specifically, __temporal dependencies__ are the influence of past values on future values. For example, today's weather can be predicted based on yesterday's weather conditions.
> __Seasonal patterns__ denote repetitive patterns that occur at regular intervals, such as daily, weekly, or yearly cycles.
> __Trend patterns__ are long-term upward or downward movements in the data.
> __Autocorrelation__ is the correlation between a time series and a lagged version of itself.
> __Multivariate correlation__ is the correlation among multiple time series.
>
> In time series forecasting, trend-seasonal decomposition is a common technique to tackle intricate temporal patterns.
> In lines 053-084 of our manuscript, we point out that __existing decomposition designs__ generally utilize two __independent branches__ to highlight seasonal and trend properties separately for final prediction.
> Considering trend-seasonal decomposition is an __untrainable linear transformation__ (i.e., moving average kernel), the decomposed trend and seasonal components __cannot__ accurately characterize the inherent dependencies of the raw time series.
>
> To intuitively understand the limitation of the seasonal-trend decomposition, we provide the trend-seasonal decomposition visualization of two channels (i.e., variates) on the Electricity and Traffic datasets.
> As seen in Figure 2 of our manuscript, the "Observed" denotes the raw time series, and the "Trend" and "Seasonal" are decomposed components from the raw time series by the moving average kernel.
> __Comparatively__, the trend and seasonal components exhibit distinct characteristics for the observed time series, where accurately capturing the inherent dependencies from the seasonal or trend components __alone is impossible__.
> Taking __channel 11__ on the Electricity dataset as an example, the long-term upward or downward movements roughly depicted by __the trend component__ of channel 11 __cannot reflect the periodic patterns__ of raw channel 11, while the repetitive patterns depicted by __the seasonal component__ of channel 11 are __not actual temporal patterns__ of channel 11.
> __Based on the observation, we believe the interactions between decomposed components are important for time series forecasting.__
>
> Benefiting from the ability to model __long-term dependencies (i.e., autocorrelation and multivariate correlation)__ by the attention module, Transformer-based methods have shown significant success in time series forecasting.
> __To learn the interactions between decomposed components, we take the attention mechanism as the basic module to connect the seasonal and trend components.__
> To avoid excessive computational overhead caused by the attention mechanism, we must consider how to obtain the multivariate correlations for different components.
> In __Figure 2__ of the manuscript, we observe that the seasonal components can better reflect the temporal patterns of observed time series, so we use the seasonal components to learn the multivariate correlation of time series data.
> Based on __Equation (1)__ of the manuscript, we can find that the seasonal component is obtained by subtracting the trend component from the original time series, so we replace the addition operation of the existing Transformer-based architecture with a subtraction mechanism.
> In __lines 243-269__ of our manuscript, we provide a rationality analysis for our dual-stream framework.
>
>
> __In summary__, by delving into the trend-seasonal decomposition process and combining the attention mechanism's remarkable ability to learn long-term dependencies, our proposed Transformer-based dual-stream interaction framework is __traceable rather than heuristic__.
> Based on our interactive architecture, we can learn __inherent dependencies beyond the multivariate correlation__ to improve time series forecasting.
> Regarding the __design choices__, we acknowledge that some may initially appear ad-hoc.
> However, we would like to clarify that these choices were made based on a __combination__ of empirical evidence and theoretical considerations. While some decisions were indeed informed by preliminary experiments, they were also __guided by established principles__ in the field.

---

> ### Author Response · Authors · 2024-11-25
>
> Dear Reviewer pkJh,
>
> Sorry to bother you again.
>
> Given the importance of timely communication and the rebuttal phase nearing the end, we would like to know if we have addressed your concerns. If you have any remaining questions, please let us know. We are looking forward to your valuable reply.
>
> Thank you for your efforts in our paper.
>
> Best regards,
>
> Authors

---

> > ### Comment · Reviewer_pkJh · 2024-11-26
> > **Thanks**
> >
> > Thanks for the explanation. I have increased score.

---

> > > ### Author Response · Authors · 2024-11-26
> > >
> > > Once again, we sincerely thank you for your prompt response.
> > >
> > > In addition to the above statements, we provide more visual results in __Figures 9-12 of the revised manuscript__ to help you understand the seasonal and trend components learned by our method.
> > >
> > > We hope these visualizations will further enhance your understanding and appreciation of our work.

---

> > > > ### Author Response · Authors · 2024-11-30
> > > >
> > > > Dear Reviewer pkJh,
> > > >
> > > > As the author/reviewer discussion will close soon, we would like to know if our response has addressed your concerns and questions. If you have any further concerns or suggestions for the paper or our rebuttal, please let us know. We would be happy to engage in further discussion.
> > > >
> > > > Thank you again for the time and effort you dedicated to reviewing this work.
> > > >
> > > > Sincerely,
> > > >
> > > > Authors

---

### Official Review · Reviewer_ioRG · 2024-11-03

**Soundness:** 3
**Presentation:** 4
**Contribution:** 2
**Rating:** 5
**Confidence:** 4

**Summary:**

The paper proposed the TwinsFormer architecture for time series forecasting. TwinsFormer decomposes the observed time series into trend and seasonal components and designs a dual-branch Transformer architecture that models their interaction at each layer. For the seasonal branch, the author applies an FFN + attention architecture that is similar to iTransformer. The only difference is that author subtracts the attention output from the input embeddings to capture the seasonal variations. For the trend branch, the autor adopts a convolution-based structure called Interactive Module (IM). The author compared TwinsFormer with a few baselines on 9 datasets and also conducted ablation study on the design choices. Experiments show that TwinsFormer consistently outperforms baselines. In addition, the author applied the dual-branch design in other transformer-based time-series forecasting models and observed consistent performance boost.

**Strengths:**

The paper is well-written and it is easy to understand the design and motivation of TwinTransformer. The author also conducted experiments and ablation studies on a variety of datasets to verify the conclusions.

**Weaknesses:**

The contribution is marginal because several previous papers (e.g., DESTformer: A Transformer Based on Explicit Seasonal–Trend Decomposition for Long-Term Series Forecasting, https://www.mdpi.com/2076-3417/13/18/10505) have tried to decompose the Transformer into trend and seasonal component and fuse the encoded features at each layer. Actually, if we closely insepct Table 3, Variant #2 performs similarly as TwinsFormer. This may indicate that part of the dual-branch design (e.g., the choice of using Interactive Module IM) is not that beneficial compared with the high-level choice of separately modeling the seasonal and trend features. In addition, the performance boost of TwinsFormer may come from adopting iTransformer in the seasonal branch. We can compare the iTransformer column and TwinsTransformer column in Table 1, 2, 6, 7 and notice that iTransformer performs well in most benchmarks.

**Questions:**

What's the variance of the performance numbers in Table 3? Are they really significant?

---

> ### Author Response · Authors · 2024-11-21
> **Response to Reviewer ioRG (Part 1)**
>
> Thank you for your time in reviewing our paper and the detailed comments. The followings are our responses to your concerns and questions:
>
> __W1__: Difference to DESTformer.
>
> DESTformer is an interesting work that fuses the encoded features of decomposed components at each layer of the model via the addition operation. However, our method and DESTformer are completely different. 1) Since we encode the historical time series of each variate into an embedding, our seasonal-trend decomposition acts in the observed time series space rather than the embedding space of DESTformer. 2) DESTformer is designed to alleviate the information utilization bottleneck of existing decomposition-based transformers, while TwinsFormer is the first attempt to explore the interactions between decomposed components to break through the limitation of the seasonal-trend decomposition.
>
> __W2__: Different performance to iTransformer.
>
> In model variate ② of Table 3, we feed trend components to the seasonal branch and feed seasonal components to the trend branch.
> The similar performance to TwinsFormer only indicates that swapping inputs for different branches has relatively little impact on model performance.
> To further illustrate the role of our interactive module in our framework, we disable our interactive module in the trend branch as the model variate ⑧.
> Specifically, we feed the seasonal components obtained by the moving average kernel to the seasonal branch and treat the trend components obtained by the moving average kernel as the output of the trend branch.
> The results of ⑧ on ECL, Traffic, PEMS03, and PEMS07 datasets are listed below:
> | TwinsFormer |      ECL     |    Traffic   |    PEMS03    |    PEMS07    |
> |:-------:|:------------:|:------------:|:------------:|:------------:|
> |    96   | 0.139\|0.233 | 0.382\|0.260 | 0.065\|0.169 | 0.060\|0.158 |
> |   192   | 0.158\|0.252 | 0.392\|0.267 | 0.086\|0.196 | 0.079\|0.181 |
> |   336   | 0.172\|0.267 | 0.410\|0.276 | 0.121\|0.234 | 0.104\|0.209 |
> |   720   | 0.200\|0.293 | 0.442\|0.292 | 0.165\|0.276 | 0.132\|0.236 |
> |   avg   | 0.167\|0.262 | 0.406\|0.273 | 0.109\|0.219 | 0.094\|0.196 |
>
> | ⑧  |      ECL     |    Traffic   |    PEMS03    |    PEMS07    |
> |:-------:|:------------:|:------------:|:------------:|:------------:|
> |    96   | 0.156\|0.248 | 0.396\|0.268 | 0.071\|0.177 | 0.062\|0.162 |
> |   192   | 0.172\|0.264 | 0.412\|0.275 | 0.099\|0.211 | 0.085\|0.191 |
> |   336   | 0.188\|0.278 | 0.423\|0.292 | 0.169\|0.282 | 0.170\|0.283 |
> |   720   | 0.225\|0.325 | 0.449\|0.308 | 0.197\|0.306 | 0.197\|0.301 |
> |   avg   | 0.185\|0.279 | 0.420\|0.286 | 0.134\|0.244 | 0.129\|0.234 |
>
> Without the interactive module,  the performance degradations of model variate ⑧ are obvious, even worse than iTransformer.
> These degradations indicate that despite iTransformer providing a good backbone for our approach, our interactive module plays a key role in further improving forecasting performance.
>
> __Q1__: Considering that an untrainable linear transformation (i.e., the moving average kernel) cannot reflect the non-linear pattern of the observed time series, we propose a Transformer-based interaction framework to better capture the intrinsic dependencies for time series forecasting.
> __Since we utilize the model to learn the interactions between decomposed components for time series forecasting, it is significant to analyze the impact of the inputs and operations of key modules on the model performance__.
> The results in Table 3 are obtained with the same fixed random seed, which means the result is the same in each run under the same hyperparameters.
> Since the variance is too small, we compute the standard deviation of ablation studies with five random seeds as below:

---

> ### Author Response · Authors · 2024-11-21
> **Response to Reviewer ioRG (Part 2)**
>
> |    |      ECL     |    Traffic   |    PEMS03    |    PEMS07    |
> |:-------:|:------------:|:------------:|:------------:|:------------:|
> |Ours|0.167$\pm$0.001\|0.262$\pm$0.001|0.406$\pm$0.001\|0.273$\pm$0.002|0.109$\pm$0.002\|0.219$\pm$0.001|0.094$\pm$0.002\|0.196 $\pm$ 0.001|
> ①|0.176$\pm$0.001\|0.272$\pm$0.002|0.417$\pm$0.002\|0.282$\pm$0.001|0.116$\pm$0.001\|0.226$\pm$0.002|0.102$\pm$0.001\|0.204$\pm$0.002|
> |②|0.172$\pm$0.001\|0.265$\pm$0.001|0.413$\pm$0.001 \|0.277$\pm$0.001|0.114$\pm$0.002\|0.224$\pm$0.001|0.101$\pm$0.002 \|0.204$\pm$0.002|
> | ③ |0.180$\pm$0.002\|0.275$\pm$0.001|0.416$\pm$0.002\|0.283$\pm$0.001|0.118$\pm$0.001\|0.228$\pm$0.002| 0.102$\pm$0.002\|0.207$\pm$0.001|
> |④ |0.185$\pm$0.001\|0.278$\pm$0.002|0.418$\pm$0.001\|0.283$\pm$0.002|0.122$\pm$0.001\|0.232$\pm$0.001| 0.105$\pm$0.001\|0.210$\pm$0.002|
> |⑤ |0.183$\pm$0.002\|0.277$\pm$0.001|0.413$\pm$0.001\|0.278$\pm$0.001|0.118$\pm$0.001\|0.229$\pm$0.001|  0.103$\pm$0.001\|0.208$\pm$0.001|
> |⑥ |0.176$\pm$0.001\|0.271$\pm$0.001|0.412$\pm$0.001\|0.281$\pm$0.001|0.121$\pm$0.002\|0.228$\pm$0.002| 0.104$\pm$0.002\|0.210$\pm$0.001|
> | ⑦ | 0.176$\pm$0.002\|0.268$\pm$0.001|0.413$\pm$0.002\|0.278$\pm$0.001|0.118$\pm$0.001\|0.228$\pm$0.001|  0.107$\pm$0.001\|0.215$\pm$0.002|
> |⑧|0.185$\pm$0.002\|0.279$\pm$0.002|0.420$\pm$0.002\|0.286$\pm$0.001|0.134$\pm$0.002\|0.244$\pm$0.001|  0.129$\pm$0.002\|0.234$\pm$0.001 |
>
> These results exhibit that the performance of TwinsFormer is stable in ablation studies.

---

> ### Author Response · Authors · 2024-11-25
>
> Dear Reviewer ioRG,
>
> Sorry to bother you again.
>
> Given the importance of timely communication and the rebuttal phase nearing the end, we would like to know if we have addressed your concerns. If you have any remaining questions, please let us know. We are looking forward to your valuable reply.
>
> Thank you for your efforts in our paper.
>
> Best regards,
>
> Authors

---

> > ### Author Response · Authors · 2024-11-30
> >
> > Dear Reviewer ioRG,
> >
> > As the author/reviewer discussion will close soon, we would like to know if our response has addressed your concerns and questions. If you have any further concerns or suggestions for the paper or our rebuttal, please let us know. We would be happy to engage in further discussion.
> >
> > Thank you again for the time and effort you dedicated to reviewing this work.
> >
> > Sincerely,
> >
> > Authors

---

> > > ### Comment · Reviewer_ioRG · 2024-11-30
> > >
> > > Thanks for the rebuttal. I still think the method lacks novelty and won't change the score.

---

> ### Author Response · Authors · 2024-12-01
>
> Anyway, thank you for your prompt reply.
>
> In this paper, we __delve into__ the trend-seasonal decomposition and find that the seasonal and trend components obtained by the trend-seasonal decomposition __cannot accurately capture__ the inherent dependencies of the time series due to the __untrainable linear transformation__ (i.e., the moving average kernel).
>
> Therefore, we propose a __first Transformer-based framework__ that explicitly explores inherent dependencies by __learning implicit and progressive interactions__ between __different components__ for time series forecasting.
>
> Concretely, we design an interactive module around the attention module and the feed-forward network to __learn more robust and reliable decomposed representations__ based on residual and interactive learning.
>
> Technically, we __only utilize__ the trend-seasonal decomposition to __initialize__ our seasonal and trend components, and our dual-stream interaction framework can enhance the __transformation and decoupling__ of seasonal and trend.
>
> To better understand the learning process of the seasonal and trend components, we provide more visualization results in __Figures 10-12 of the revised manuscript__.
>
> Experimentally, TwinsFormer achieves __state-of-the-art performances__ in both long-term and short-term forecasting tasks, and is __plug-and-play__ into existing Transformer-based methods with __negligible extra computational overhead__.
>
> We understand your concern regarding the novelty of our method. To better address this issue, we would greatly appreciate it if you could provide more detailed feedback on the specific aspects of our approach that you find lacking in novelty. Your insights would help us understand how we can further enhance the originality and impact of our work.
>
> We look forward to your further feedback and are committed to addressing any concerns raised by you.

---

### Official Review · Reviewer_AH3t · 2024-11-05

**Soundness:** 3
**Presentation:** 3
**Contribution:** 3
**Rating:** 6
**Confidence:** 3

**Summary:**

This paper proposed TwinsFormer  for time series forecasting that uses dual streams to interactively process trend and seasonal components, enhancing dependency learning between them. By integrating these interactions within the attention and feed-forward layers, TwinsFormer improves forecasting accuracy without adding significant computational cost.

**Strengths:**

(1) The proposed TwinsBlock is novel and well justified by rationality analysis, ablation and visualization analysis.

(2) The proposed TwinsFormer demonstrates better performance than  baseline models  in most datasets,

(3) The paper is well-organized and easy to follow.

**Weaknesses:**

(1) The improvements of TwinsFormer over baselines in the Main results (Tables 1 and 2) are quite marginal.  Generally, the improvement over the second-best baseline is less than **0.02** for MSE and MAE metrics.

(2) Given the marginal improvements over baselines shown in Tables 1 and 2, it is essential to include the variance of repeated experiments for a more accurate assessment.

(3)  In Equation (9), the authors justify their design with the assumption of "replacing [.] with +". Given that '+' appears to be a more appropriate choice based on your analysis, why not simply sum up \( X_t \), \( g(X_s) \), and \( h(X_s, g(X_s)) \)? Is there any evidence to support concatenation as the optimal design choice?

(4) It seems that all evaluations are currently conducted on multivariate forecasting tasks. Including evaluations on univariate forecasting tasks would provide a more comprehensive assessment for TwinsFormer.

**Questions:**

(1) Is the improvement in Tables 1 and 2 statistically significant? Could you report the confidence level using a statistical hypothesis test ,e.g. T-test?

(2) Does the proposed TwinFormer still work well for univariate forecasting tasks?

---

> ### Author Response · Authors · 2024-11-21
> **Response to Reviewer AH3t (Part 1)**
>
> Many thanks to Reviewer AH3t for providing a detailed review and insightful questions.  We address your concerns here:
>
> __W1__: The metrics used in time series forecasting, Mean Squared Error (MSE) and Mean Absolute Error (MAE), are highly sensitive to small changes, especially when the baseline models are already performing at a high level.
> A difference of less than 0.02 in these metrics can make a significant improvement in practical applications, particularly in domains where even minor errors can have substantial consequences, such as financial and traffic forecasting.
> Furthermore, All experimental results were obtained using the same random seed, meaning that running the experiments with the same parameters consistently yields identical results.
> Additionally, the results in Tables 1 and 2 are __the average results__ of four different forecasting lengths, and we provide the full experimental results in __Tables 6-7__ of the manuscript.
>
> __W2\&Q1__: To ensure the fairness of the experiment, all experimental results are obtained by running under the same random seed.
> Meanwhile, we reported the __standard deviations__ of TwinsFormer performance in __Table 8__ of the manuscript.
> We also provide statistical tests on ECL and all subsets of ETT and PEMS with five runs of random seeds.
> The performance is averaged from four prediction lengths.
> The standard deviations and T-test statistics (__MSE|MAE__) of TwinsFormer and iTransformer are listed below, showing that the performance is on par with the previous SOTA iTransformer within the margin of error.
> |              |                 ETT                |                 ECL                |                PEMS                |
> |:------------:|:----------------------------------:|:----------------------------------:|:----------------------------------:|
> | iTransformer | 0.383 $\pm$ 0.001 \|  0.399  $\pm$ 0.002 | 0.178 $\pm$ 0.001 \| 0.270 $\pm$ 0.001 | 0.122 $\pm$ 0.002 \| 0.224 $\pm$ 0.001 |
> | TwinsFormer | 0.372 $\pm$ 0.001 \|  0.392  $\pm$ 0.001 |  0.167$\pm$ 0.001 \| 0.262 $\pm$ 0.001 | 0.112 $\pm$ 0.001 \| 0.214 $\pm$ 0.002 |
> |    T-test    |            2.158\|2.412            |           -1.179\|-1.196           |           -0.763\|-0.787           |
>
> __W3__: In the rationality analysis of Section 3, we formulate the model to provide a formal expression to facilitate the understanding of the practical significance of our dual-stream structure and its compatibility with the decomposition design.
> Replacing concatenation with addition operations is intended to simplify the formula of the interactions between decomposed components rather than highlight which operation is the optimal design choice.
> To better illustrate the __difference between the addition and concatenation operations__, we compare the average performance (__MSE|MAE__) of the two operations on three datasets as below.
> | Type |    Weather   |      ECL     |    Traffic   |
> |:----:|:------------:|:------------:|:------------:|
> |  []  | 0.246\|0.271 | 0.167\|0.262 | 0.406\|0.273 |
> |   +  | 0.249\|0.274 | 0.172\|0.265 | 0.410\|0.280 |
>
> The results indicate that the performance of '+' is inferior to that of '[]', which may be due to the addition operation inhibiting the decoupling of seasonal and trend components.
>
> __W4\&Q2__: The performance (__MSE|MAE__) of univariate time series forecasting on all the subsets of ETT is presented below.
> As iTransformer, TimeMixer, and Crossformer do not offer training hyperparameters and results for univariate forecasting tasks, we compare our method with PatchTST, DLinear, and FEDformer.
> | ETTh1 |  TwinsFormer  |    PatchTST   |    DLinear    |   FEDformer   |
> |-------|:-------------:|:-------------:|:-------------:|:-------------:|
> | 96    | 0.057\|0.179  | 0.059\|0.189  | 0.056\|0.180  | 0.079\|0.215  |
> | 192   | 0.072\|0.210  | 0.074\|0.215  | 0.071\|0.204  | 0.104\|0.245  |
> | 336   | 0.080\|0.219  | 0.076\|0.220  | 0.098\|0.244  | 0.119\|0.270  |
> | 720   | 0.079\|0.228  | 0.087\|0.236  | 0.189\|0.359  | 0.142\|0.299  |
> | avg   | 0.072\|0.209  | 0.074\|0.215  | 0.104\|0.247  | 0.111\|0.257  |
>
> | ETTh2 |  TwinsFormer  |    PatchTST   |    DLinear    |   FEDformer   |
> |-------|:-------------:|:-------------:|:-------------:|:-------------:|
> | 96    | 0.129\|0.275  | 0.131\|0.284  | 0.131\|0.279  | 0.128\|0.271  |
> | 192   | 0.178\|0.329  | 0.171\|0.329  | 0.176\|0.329  | 0.185\|0.330  |
> | 336   | 0.210\|0.346  | 0.171\|0.336  | 0.209\|0.367  | 0.231\|0.378  |
> | 720   | 0.221\|0.378  | 0.223\|0.380  | 0.276\|0.426  | 0.278\|0.420  |
> | avg   | 0.185\|0.332  | 0.174\|0.332  | 0.198\|0.350  | 0.206\|0.350  |

---

> > ### Author Response · Authors · 2024-11-21
> > **Response to Reviewer AH3t (Part 2)**
> >
> > | ETTm1 |  TwinsFormer  |    PatchTST   |    DLinear    |   FEDformer   |
> > |-------|:-------------:|:-------------:|:-------------:|:-------------:|
> > | 96    | 0.027\|0.126  | 0.026\|0.123  | 0.028\|0.123  | 0.033\|0.140  |
> > | 192   | 0.042\|0.154  | 0.040\|0.151  | 0.045\|0.156  | 0.058\|0.186  |
> > | 336   | 0.055\|0.180  | 0.053\|0.174  | 0.061\|0.182  | 0.084\|0.231  |
> > | 720   | 0.076\|0.209  | 0.073\|0.206  | 0.080\|0.210  | 0.102\|0.250  |
> > | avg   | 0.050\|0.167  | 0.048\|0.164  | 0.054\|0.168  | 0.069\|0.202  |
> >
> > | ETTm2 |  TwinsFormer  |    PatchTST   |    DLinear    |   FEDformer   |
> > |-------|:-------------:|:-------------:|:-------------:|:-------------:|
> > | 96    | 0.064\|0.185  | 0.065\|0.187  | 0.063\|0.183  | 0.067\|0.198  |
> > | 192   | 0.095\|0.232  | 0.093\|0.231  | 0.092\|0.227  | 0.102\|0.245  |
> > | 336   | 0.123\|0.265  | 0.121\|0.266  | 0.119\|0.261  | 0.130\|0.279  |
> > | 720   | 0.170\|0.317  | 0.172\|0.322  | 0.175\|0.320  | 0.178\|0.325  |
> > | avg   | 0.113\|0.250  | 0.113\|0.252  | 0.112\|0.248  | 0.119\|0.262  |
> >
> > These experimental results show that our method can achieve competitive performance in univariate forecasting tasks.

---

> > > ### Author Response · Authors · 2024-11-30
> > >
> > > Dear Reviewer AH3t,
> > >
> > > As the author/reviewer discussion will close soon, we would like to know if our response has addressed your concerns and questions. If you have any further concerns or suggestions for the paper or our rebuttal, please let us know. We would be happy to engage in further discussion.
> > >
> > > Thank you again for the time and effort you dedicated to reviewing this work.
> > >
> > > Sincerely,
> > >
> > > Authors

---

### Official Review · Reviewer_eGh2 · 2024-11-07

**Soundness:** 2
**Presentation:** 2
**Contribution:** 2
**Rating:** 5
**Confidence:** 4

**Summary:**

This paper introduces a Transformer-based model (TwinsFormer) to improve time series forecasting by capturing long-term dependencies. The key point of the model is to enable the interaction of decomposed time series components, which is realized by employing a dual-stream approach with an attention module and a feed-forward network to strengthen dependencies between trend and seasonal parts. Experiments demonstrate that TwinsFormer outperforms previous state-of-the-art methods in long-term and short-term tasks. Further analysis reveals that the interactive strategy can be a plug-and-play module to existing Transformer-based methods.

**Strengths:**

* The motivations are well-summarized and the paper's writing is easy to follow.
* There is a good summary of the differences from previous works, such as the decomposition of observed time series rather than the time series embeddings, which I think is reasonable.

**Weaknesses:**

* This work is not innovative in the decomposition, which I think is worth further exploration. The claim "TwinsFormer is the first attempt to consider interactions between decomposed components on Transformer for time series forecasting" can be overstated (e.g., TimeMixer).
* How about the results of Table 1 if using a longer lookback length? I think the lookback window of lookback=96 is too short to obtain the robustly decomposed trend and seasonal components. These results would greatly influence my evaluation of this work.
* Whether the ablation in Table 4 can further improve the performance on more recent Transformer-based forecasters, for example, PatchTST, iTransformer, and Crossformer.
* Does the model adopt channel independence? Are the interactions between decomposed components carried out independently within the channel or among multiple channels?
* The efficient version of TwinsFormer is trained with 20% variates and prediction for all variates, which is the same as iTransformer. I don't think it is an innovative part to place these experiments. Similarly, other model analyses basically follow the previous work (such as TimeMixer), so I think Sections 4.2 and 4.3 lack originality and do not provide enough insight to readers. It should undergo a major revision to delve more into the perspective of time series decomposition.

**Questions:**

* It seems that the model only decomposed input series at the beginning. Could you verify that the subsequent TwinsBlock can still adequately deal with trend and seasonal components? Does it matter if you randomly swap the Treand-Seasonal input for a TwinsBlock in the middle of the model?
* Can the author provide more concrete showcases: How does Twinsformer cope with the intricate interactions between decomposed components to precisely unravel the inherent dependencies as mentioned in Figure 2?
* Why does Figure 1 compute MAE instead of MSE?

Updated after rebuttal: Thanks for the responses. My concern is partially addressed, and I have adjusted my score to 5.

---

> ### Author Response · Authors · 2024-11-21
> **Response to Reviewer eGh2 (Part 1)**
>
> Thanks for reviewing our paper.  The following are our responses to your specific questions and concerns:
>
> __W1__: In the manuscript, we highlight the __interactions between decomposed components (i.e., seasonal and trend components)__ rather than the new design for the decomposition.
> Specifically, we employ the moving average kernel to decompose the time series data into seasonal and trend components, and then design a Transformer-based dual-stream framework to make different components learn from each other and capture better inherent dependencies for the final forecasting.
> In addition, we supplement the experimental results of wavelet and non-linear decompositions as described by Reviewer WMfY, which illustrates the compatibility of our dual-stream framework for more decomposition designs.
> As an __MLP-based__ framework, __TimeMixer__ introduced the decomposable multiscale mixing module to learn the __interactions among different scale seasonal (or trend) components__ rather than the __interactions between seasonal and trend components__.
> Existing decomposition designs treat the seasonal and trend components as two unrelated variables, ignoring the interactions between the decomposed components.
> Therefore, __TwinsFormer is the first attempt to consider interactions between decomposed components on Transformer__ for time series forecasting.
>
> __W2:__: Indeed, different lookback window lengths affect the robustness of the decomposed trend and seasonal components.
> We __analyzed the lookback length sensitivity__ in lines 408-427 of our manuscript, and __Figure 4__ clearly illustrates that our TwinsFormer can effectively capture inherent dependencies from a longer lookback window.
>
> __W3__: Generally, the most important module of Transformer-based forecasters is the attention mechanism.
> In Table 4 of our manuscript, we mainly analyze the impact of different attention mechanisms on the performance of our framework.
> For __PatchTST__ and __Crossformer__, the time series fed to their attention modules is obtained by __patching and segmentation operations__, which cannot capture the seasonal and trend characteristics of the time series.
> As for __iTranformer__, its success lies in learning inherent dependencies from the dimension of variates rather than exploring new attention mechanisms.
> Without the decomposition and interactive modules, our backbone is consistent with iTransformer.
> Therefore, we did not analyze the attention compatibility and performance promotion on PatchTST, iTransformer, and Crossformer in Table 4.
>
> __W4__: In our work, we adopt __channel independence__ to decompose the lookback length of each channel into seasonal and trend components, and then encode the decomposed components into corresponding embeddings.
> We utilize the attention mechanism in the seasonal branch to learn the multivariate correlation among different seasonal embeddings.
> Meanwhile, we treat the residual seasonal signals obtained by the subtraction mechanism as the supervision information to perform interactive learning with the trend embeddings in the trend branch.
> In other words, our interactions between decomposed components among __multiple channels__.
>
> __W5__:  Both iTransformer and TimeMixer are excellent works, and __we adopted some of their experimental formats because their approaches to model analysis are highly representative and widely recognized__ in the field.
> Moreover, our dual-stream Transformer-based framework incorporates both decomposition design and a plug-and-play interactive module, necessitating an exploration of the individual components' roles and the generalization capabilities of our module.
> While the analysis format may resemble existing methods, our primary focus is on the __unique contributions and effectiveness__ of our dual-stream Transformer-based framework.
>
> __Q1__: We regard the decomposed components obtained by the moving average kernel as the initial decomposed components, and then feed the encoded seasonal and trend embeddings to the TwinsBlock.
> To verify our TwinsBlock can still adequately deal with the trend and seasonal components, we supplement additional visualization results in __Figure 10 of the revised paper__.
> In Table 3 of the manuscript, we conducted the ablation study on swapping seasonal and trend components (i.e. __the model variant__ ②), and the results show that __feeding the seasonal components to the attention module is more beneficial for model performance__.
> Recall __Figure__ 2 of the manuscript, the trend components are very smooth over a certain time interval, while the seasonal components can better reflect the temporal patterns of the observed time series, which also __suggests that the seasonal components can better represent the multivariate correlations__.
> Thus, we do not need to consider randomly swapping the trend-seasonal inputs to a TwinsBlock in the middle of the model.

---

> > ### Author Response · Authors · 2024-11-21
> > **Response to Reviewer eGh2 (Part 2)**
> >
> > __Q2__: As seen in __Figure 11-12 of the revised paper__, we provide concrete showcases on the ECL and Traffic benchmarks, clearly illustrating the distinctions between the decomposed components learned by the moving average kernel and those derived from our interaction mechanisms.
> >
> > __Q3__: In Tables 1 and 2 of the manuscript, we provide the full average performance among all the benchmarks in both MSE and MAE.
> > These results show that TwinsFormer performs better in MAE for all the benchmarks, so we plot the MAE results instead of MSE results in Figure 1 to better __highlight the superiority__ of our method.

---

> ### Author Response · Authors · 2024-11-25
>
> Dear Reviewer eGh2,
>
> Sorry to bother you again.
>
> Given the importance of timely communication and the rebuttal phase nearing the end, we would like to know if we have addressed your concerns. If you have any remaining questions, please let us know. We are looking forward to your valuable reply.
>
> Thank you for your efforts in our paper.
>
> Sincerely,
>
> Authors

---

> > ### Author Response · Authors · 2024-11-30
> >
> > Dear Reviewer eGh2,
> >
> > As the author/reviewer discussion will close soon, we would like to know if our response has addressed your concerns and questions. If you have any further concerns or suggestions for the paper or our rebuttal, please let us know. We would be happy to engage in further discussion.
> >
> > Thank you again for the time and effort you dedicated to reviewing this work.
> >
> > Sincerely,
> >
> > Authors

---

### Official Review · Reviewer_WMfY · 2024-11-10

**Soundness:** 3
**Presentation:** 4
**Contribution:** 4
**Rating:** 8
**Confidence:** 4

**Summary:**

The paper presents TwinsFormer, a Transformer-based framework designed for time series forecasting. The primary innovation lies in its ability to model interactions between the trend and seasonal components of time series data via an interactive dual-stream design.

**Strengths:**

+ The proposed framework addresses the challenge of accurately capturing the intricate temporal dependencies between trend and seasonal components in time series forecasting. The dual-stream design and interaction mechanism are novel, offering an efficient way to learn the dependencies between these components while maintaining computational efficiency.
+ The paper provides a clear and rigorous explanation of why traditional independent decomposition models (which separate trend and seasonal components) might miss crucial interactions.
+ The authors highlight that their interactive strategy can be applied to existing Transformer-based architectures with minimal computational overhead.
+ Extensive experiments across 13 real-world datasets, including both short-term and long-term forecasting scenarios, demonstrate the effectiveness of TwinsFormer. The paper shows that it outperforms several state-of-the-art models, such as iTransformer, TimeMixer, and Crossformer, across a variety of metrics (MSE, MAE).

**Weaknesses:**

+  The paper uses a simple moving average kernel for decomposing the time series into trend and seasonal components, which assumes that the trend and seasonal components are captured well by linear operations. Could more sophisticated decomposition techniques (such as wavelet or non-linear methods) potentially offer better results?
+ The paper doesn’t provide detailed analysis regarding the sensitivity of the model’s performance to hyperparameters, especially those related to the decomposition mechanism (such as kernel size).
+ The dual-stream design, although innovative, introduces significant architectural complexity.  While the paper mentions that the model is computationally efficient, it does not provide a comprehensive analysis of the computational cost for both training and inference across different model sizes (e.g., small, medium, and large).
+ The paper introduces a subtraction mechanism in the seasonal branch of the model to eliminate redundant information. How about other operations like concatenation or addition?

**Questions:**

Check Weaknesses

---

> ### Author Response · Authors · 2024-11-21
> **Response to Reviewer  WMfY**
>
> Many thanks to Reviewer WMfY for providing thorough and insightful comments. The following are our responses to your specific questions and concerns.
>
> __W1__: Thanks for your scientific rigor. We compare the performance (__MSE|MAE__) of  the moving average kernel (__MAK__) with Wavelet (__Wav__) and Non-linear (__Non__) decomposition designs on three benchmarks below:
> | MAK |    Weather    |      ECL      |    Traffic    |
> |:--------:|:-------------:|:-------------:|:-------------:|
> |    96    |  0.161\|0.201 | 0.139\|0.233  |  0.382\|0.260 |
> |    192   |  0.211\|0.248 | 0.158\|0.252  | 0.392\|0.267  |
> |    336   | 0.266\|0.291  | 0.172\|0.267  | 0.410\|0.276  |
> |    720   | 0.347\|0.343  | 0.200\|0.293  | 0.442\|0.292  |
> |    avg   | 0.246\|0.271  | 0.167\|0.262  | 0.406\|0.273  |
>
> | Wav |    Weather    |      ECL      |    Traffic    |
> |:--------:|:-------------:|:-------------:|:-------------:|
> |    96    | 0.160\|0.200  | 0.138\|0.232  | 0.381\|0.260  |
> |    192   | 0.210\|0.248  | 0.157\|0.251  | 0.390\|0.265  |
> |    336   | 0.265\|0.290  | 0.171\|0.266  | 0.409\|0.275  |
> |    720   | 0.345\|0.345  | 0.200\|0.292  | 0.438\|0.290  |
> |    avg   | 0.245\|0.271  | 0.167\|0.260  | 0.405\|0.273  |
>
> | Non |    Weather    |      ECL      |    Traffic    |
> |:--------:|:-------------:|:-------------:|:-------------:|
> |    96    | 0.158\|0.198  | 0.136\|0.230  | 0.380\|0.258  |
> |    192   | 0.207\|0.246  | 0.155\|0.250  | 0.388\|0.263  |
> |    336   | 0.263\|0.287  | 0.171\|0.267  | 0.405\|0.270  |
> |    720   | 0.343\|0.342  | 0.199\|0.293  | 0.435\|0.286  |
> |    avg   | 0.243\|0.268  | 0.165\|0.260  | 0.402\|0.269  |
>
> These results show that our method has good compatibility with various decomposition techniques.
>
> __W2__: Thanks for your valuable comments. Since the existing decomposition designs uniformly use the kernel size of 25 by default, we ignore the decomposition mechanism's hyperparameters. We provide the results (__MSE|MAE__) related to the kernel size (__KS__) and embedding dimension (__Dim__) on ECL and Traffic below:
> | KS |      ECL      |    Traffic    |
> |:--:|:-------------:|:-------------:|
> |  5 | 0.139\|0.234  | 0.385\|0.263  |
> | 15 | 0.138\|0.233  | 0.382\|0.261  |
> | 25 | 0.139\|0.233  | 0.382\|0.260  |
> | 35 | 0.138\|0.233  | 0.383\|0.261  |
> | 45 | 0.139\|0.233  | 0.383\|0.261  |
> | 55 | 0.140\|0.234  | 0.381\|0.260  |
> | 65 | 0.139\|0.233  | 0.385\|0.262  |
> | 75 | 0.140\|0.234  | 0.383\|0.261  |
>
> |  Dim |      ECL      |    Traffic    |
> |:----:|:-------------:|:-------------:|
> |  128 | 0.152\|0.246  | 0.412\|0.284  |
> |  256 | 0.142\|0.237  | 0.394\|0.268  |
> |  512 | 0.139\|0.233  | 0.382\|0.260  |
> | 1024 | 0.139\|0.233  | 0.381\|0.260  |
>
> __W3__: To comprehensively analyze the computational cost, we treat TwinsFormer with __20%__, __60%__, and __100%__ variates as small, medium, and large model sizes, respectively.
> |       Models       | Training Times (s/Epoch) | Inference Times (s) | GPU (GB) | Paramenter (MB) | FLOPs (GB) |
> |:------------------:|:------------------------------:|:-------------------------:|:---------------------:|:---------------:|:----------:|
> |    iTransformer    |             52.25              |           14.28           |         7.50          |      6.11       |    8.63    |
> |      PatchTST      |             175.36             |           22.45           |          9.95         |      3.58       |   33.52    |
> |     Crossformer    |             283.47             |           66.19           |         11.77         |      46.90      |   229.09   |
> | Ours (100%) |             84.32              |           20.23           |         8.30          |      8.46       |   30.46    |
> |  Ours (60%) |             47.71              |           15.27           |         4.17          |      5.39       |   13.21    |
> |  Ours (20%) |             11.24              |           10.33           |         1.66          |      3.21       |    2.86    |
>
> __We can observe that our TwinsFormer requires less memory and runs faster than other Transformer-based models.__
>
> __W4__: According to __Equation (1)__ of the manuscript, we can find that the seasonal components are obtained by subtracting the trend components from the original time series in the trend-seasonal decomposition.
> To better learn the interactions between seasonal and trend components, we believe that the subtraction mechanism is more in line with the principle of trend-seasonal decomposition and further facilitates the decoupling of seasonal and trend components.
> Since __the concatenation__ will change the channel dimension of the encoded features and bring additional computational overhead, existing Transformer-based models use the addition to implement the skip connection by default.
> Therefore, Transformer-based methods do not use the concatenation to implement the skip connection.
> In __Table 3__ of the manuscript, the model variant ③ shows the __performance__ of replacing the subtraction mechanism with __the addition__.

---

> > ### Comment · Reviewer_WMfY · 2024-11-26
> >
> > Thanks for your responses. I keep my score as 8.

---

> > > ### Author Response · Authors · 2024-11-27
> > >
> > > Thank you very much for your feedback. We sincerely appreciate your valuable comments and the recognition you have given to our work.

---

### Meta-Review · Area_Chair_vkrf · 2024-12-20

**Metareview:**

This paper introduces TwinsFormer, a Transformer-based model for time-series forecasting that emphasizes modeling interactions between seasonal and trend components. Given the limited and relatively inactive discussion, AC has carefully reconsidered the paper’s contributions.

AC finds the primary shortcoming is the lack of in-depth analysis (beyond Figure 2) from the perspective of time series decomposition. The paper does not sufficiently explain why interactions between seasonal and trend components are crucial, nor how ignoring these interactions leads to inaccuracies in current forecasting models. As Reviewer eGh2 suggests,  this can not be achieved by merely replicating analysis experiments following previous literature. This gap in motivation and clarity may also explain why several reviewer concerns remain unresolved. For instance, Reviewer pkJh found the concept of “inherent dependencies” too vague without more detailed exposition.

In conclusion, the performance gains are undoubtedly acknowledged. We believe that this paper can be accepted if further providing robust justification of both its results and its underlying motivations.

**Additional Comments On Reviewer Discussion:**

Three reviewers (eGh2, ioRG, pkJh) questioned the novelty of the approach. Although the authors responded, they did not adequately address these concerns or convince the reviewers to recommend acceptance (the final scores are 8/6/5/5/5).

---

### Decision · Program_Chairs · 2025-01-22

Reject